# A *Vaccinia*-based system for directed evolution of GPCRs in mammalian cells

Christoph Klenk [1] ✉, Maria Scrivens[2], Anina Niederer [1], Shuying Shi[2], Loretta Mueller[2], Elaine Gersz [2], Maurice Zauderer [2], Ernest S. Smith[2], Ralf Strohner[3] & Andreas Plückthun [1] ✉

Directed evolution in bacterial or yeast display systems has been successfully used to improve stability and expression of G protein-coupled receptors for structural and biophysical studies. Yet, several receptors cannot be tackled in microbial systems due to their complex molecular composition or unfavorable ligand properties. Here, we report an approach to evolve G protein-coupled receptors in mammalian cells. To achieve clonality and uniform expression, we develop a viral transduction system based on *Vaccinia* virus. By rational design of synthetic DNA libraries, we first evolve neurotensin receptor 1 for high stability and expression. Second, we demonstrate that receptors with complex molecular architectures and large ligands, such as the parathyroid hormone 1 receptor, can be readily evolved. Importantly, functional receptor properties can now be evolved in the presence of the mammalian signaling environment, resulting in receptor variants exhibiting increased allosteric coupling between the ligand binding site and the G protein interface. Our approach thus provides insights into the intricate molecular interplay required for GPCR activation.

G protein-coupled receptors (GPCRs) comprise the largest family of integral membrane proteins and are involved in regulating many biological processes[1]. GPCRs sense a large variety of ligands, ranging from small molecules to large proteins, at their extracellular side. To exert their function, GPCRs sample a continuum of conformational states, resulting in levels of no activity to maximal activity. Ligands stabilize distinct conformations that either keep the receptor in an inactive state (in the case of inverse agonists) or active state (in the case of agonists)[2]. The conformational change induced by an agonist at the ligand binding pocket is transmitted to the intracellular surface of the receptor and results in activation of heterotrimeric G proteins by exchange of GDP for GTP in the Gα subunit. As a consequence, Gα and Gβγ subunits dissociate and can activate downstream signaling processes[3,4]. Given their broad physiological relevance, GPCRs are important drug targets[1], and understanding the structure-function mechanisms leading to GPCR activation are critical for future drug development. While conformational flexibility is essential for proper receptor function, it is an obstacle to experimentally determine

protein structures. This has been the motivation of substantial engineering efforts to make GPCRs amenable to structural studies.

We have previously devised several strategies using directed evolution to improve the biophysical properties of GPCRs[5–8]. Directed evolution aims to accelerate the process of natural evolution, i.e., the enrichment of desirable traits that are encoded in a pool of genetic variants, by means of iterative rounds of gene diversification and selection. Thereby, protein function can be altered or created de novo, and thus, directed evolution has become a powerful method to develop new molecular tools and therapeutics, yet mostly applied to soluble proteins. To extend this methodology to GPCRs, microbial organisms such as bacteria and yeast were used. By transformation with plasmids, they can take up, on average, one copy, such that after the plasmid establishes its copy number, each microbial cell is still "clonal". Expression of GPCRs in functional form in the plasma membrane of yeast or the inner membrane of *E. coli* has been shown[5,8,9], and thus, a strict coupling of phenotype and genotype is guaranteed which is a key requirement for directed evolution. Moreover, high

[1]Department of Biochemistry, University of Zurich, Winterthurerstrasse 190, CH-8057 Zurich, Switzerland. [2]Vaccinex, Inc., 1895 Mt. Hope Avenue, Rochester, New York 14620 NY, USA. [3]MorphoSys AG, Semmelweisstr. 7, 82152 Planegg, Germany. ✉e-mail: c.klenk@bioc.uzh.ch; plueckthun@bioc.uzh.ch

transformation rates can be achieved in microbes to efficiently represent diverse gene libraries. Using selection driven by the number of ligand-binding and thus functional receptors in the membrane, a variety of well expressing and stable GPCR variants were generated that were instrumental for structural studies and drug discovery projects[10–17].

While handling efficiency and library generation in bacteria and yeast are well suited, some fundamental limitations accompany microbial expression systems regarding their applicability for membrane proteins. The plasma membrane of microbes is typically shielded by an outer membrane or cell wall such that the ligand-binding site of receptors is not readily accessible from the extracellular side, especially for larger ligands. Efficient permeabilization protocols have been devised to render these outer barriers permeable for small molecules and short peptide ligands[5,6,8], yet bulkier molecules such as peptide hormones or proteins may not easily reach their binding site at the receptor. Also, many GPCRs are highly toxic to bacteria and yeast cells, which limits the number of potential targets to be evolved, as a minimum of basal expression is required for selection. Finally, receptor function, i.e., the potential to signal to downstream effectors, cannot easily be addressed in microbial systems because endogenous signaling cascades are missing completely (in the case of bacteria) or are only present at very reduced functionality (in the case of yeast). Therefore, GPCR evolution in microbial systems has so far been restricted to improve receptor expression and stability, without opening yet the possibility to apply the potential of directed evolution for exploring functional aspects of GPCR signaling.

In contrast, in mammalian cells no such limitations exist, and, therefore, mammalian display systems can open new possibilities for ligand-receptor combinations. Translating the ease of creating large libraries from microbial display system to mammalian cells, however, has been challenging. Many attempts to create mammalian libraries have relied on lentiviral vectors at low multiplicities of infection[18]. They are derived from the HIV genome, and their genome gets inserted into the host genome. While this would create stable libraries, there are considerable disadvantages to this approach: maximal library diversity is limited, the point of insertion determines the expression level at least as much as the phenotype of the mutant, expression level will be low as the gene copy number is very low, and the identification of protein mutants is rather tedious, as DNA can only be obtained from single-cell PCR. Furthermore, inserted transgenes can become silenced in a rather unpredictable way. Recent approaches use CRISPR/CAS technology or dedicated integrase systems that are independent of viral vectors, however at current the library size is limited[19–21]. Other methods rely on continuous directed evolution, where a low-fidelity viral vector is used creating permanent mutagenesis during viral replication. While this overcomes the challenge of creating diverse recombinant viral libraries, such systems are hard to control and require efficient selection regimes[22,23].

Here, we present a mammalian selection system for directed evolution of GPCRs, which is based on a *Vaccinia* vector. It enables facile generation of highly diverse libraries that result in homogenous expression of GPCRs, essential for subsequent expression-guided selection. We employ this system to evolve GPCRs from class A and B and demonstrate that in combination with a rational library design it can be used for extensive manipulations of GPCR properties, ranging from the generation of highly stabilized receptor variants to modulation of receptor signaling properties.

## Results

### Poxvirus vectors for directed evolution of GPCRs in mammalian cells

Monoclonal and homogenous expression of the gene of interest are critical prerequisites for selections in directed evolution that are difficult to achieve in mammalian cells with common expression strategies such as transient transfection. To reach both equal gene dosage and clonality, the cell needs to take up a single vector molecule and replicate it to a copy number that is similar in all cells. Thus, we employed a vector system which we had recently devised to generate diverse cDNA libraries in *Vaccinia* virus[24,25]. *Vaccinia* virus has a linear DNA genome of about 180 kDa which is replicated and packaged in the cytoplasm. Therefore, gene expression from the vector is homogenous between cells, and specific recombinants can be readily recovered even from very small numbers of selected cells.

To test whether uniform GPCR expression can be achieved with the *Vaccinia* vector, we assessed the expression of several variants of neurotensin receptor 1 (NTR1). A-431 cells were infected with recombinant virus at an MOI of 1 pfu per cell, thereby ensuring an average distribution of one viral particle per cell. 16–18 h after infection cells were harvested and receptor expression was analyzed by flow cytometry using fluorescently labeled ligand NT(8–13) and compared to CHO cells that had been transiently transfected with the same receptor constructs. *Vaccinia*-infected cells exhibited homogenous expression levels with narrow distribution (Fig. 1a). Thus, a clear discrimination in receptor expression between wild-type NTR1 and NTR1-TM86V and NTR1-L5X, two variants that had previously been evolved for increased functional expression in *E. coli*[26], was possible. In contrast, expression of the receptor variants in transiently transfected cells resulted in a broad distribution of expression levels for each construct, thus reflecting the varying amounts of plasmid that had been taken up by the cells (Fig. 1a). Therefore, the *Vaccinia*-based expression system enables homogenous expression of GPCRs in mammalian cells and allows clear discrimination between variants with distinct expression levels. To create large and diverse cDNA libraries, which are required for directed evolution, a specific recombination strategy has been developed which we termed Trimolecular Recombination[24]. Here, a split *Vaccinia* genome is complemented with a third fragment carrying the gene of interest together with a selection marker. Only upon recombination of all three fragments productive viral particles are formed. As a result, the background of non-recombinant virus is close to zero, enabling construction of libraries containing millions of different *Vaccinia* recombinants[24]. Based on these results, we devised a selection platform for directed evolution of GPCRs in mammalian cells which is inspired by expression-guided evolution in bacterial and yeast cells[5,8] (Fig. 1b).

### Evolution of biophysical properties of GPCRs in mammalian cells

First, we aimed to compare the capabilities of the mammalian system to selections from previous directed evolution approaches in microbial cells. We therefore chose NTR1, as our previous work has investigated this receptor in four different microbial selection systems, which have all shown successful selection for both higher stability and higher functional expression of GPCRs[5–8]. Instead of creating a fully random library by error-prone PCR, a semi-rational approach was taken to generate a synthetic library where position and type of mutation could be readily controlled. Based on the NTR1 crystal structure (PDB ID: 4BUO)[10], residues were chosen for randomization that were most likely involved in helix-helix contacts, whereas residues facing into the lipidic membrane environment or potentially being involved in ligand or G protein interaction were excluded. In total, 94 residues were selected for randomization. Mutagenesis was restricted to a predefined set of substituents for each position, which had been derived from a phylogenetic substitution matrix including the most prevalent amino acids for each position from class A GPCRs (Supplementary Fig. 1, c.f. Supplementary Notes). To make the selection outcome comparable to previous campaigns conducted in microbial display systems, we decided to uncouple the receptor from its cognate G proteins ab initio, thereby preventing any potential influence of receptor-G protein interaction, which is possible in mammalian cells

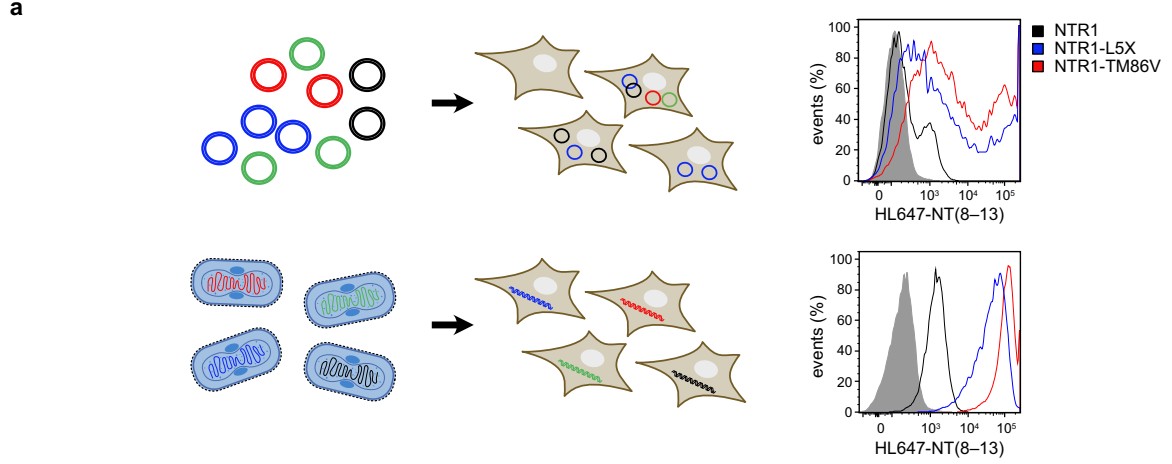

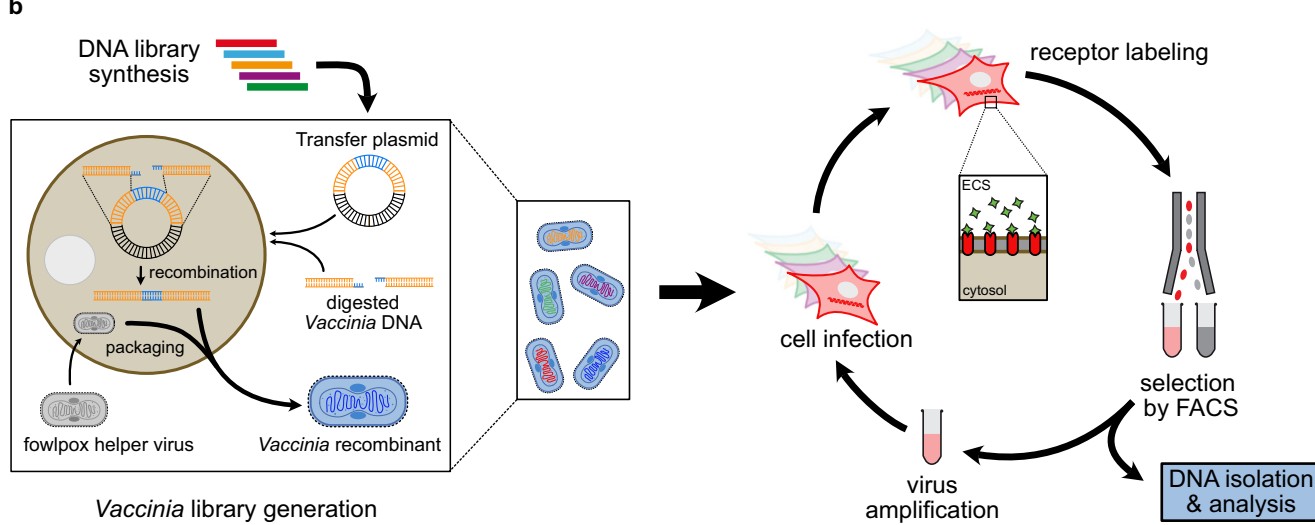

**Fig. 1 | Creating a directed evolution system for GPCRs in mammalian cells.**
**a** Comparison of GPCR expression in plasmid-transfected and *Vaccinia*-transduced mammalian cells. Transiently transfected CHO cells (upper right panel), A431 cells infected with *Vaccinia* virus constructs (lower right panel). Negative control (gray shaded), NTR1 (black line), NTR1-TM86V (red line) and NTR1-L5X (blue line) are shown. **b** Workflow for directed evolution in mammalian cells using Vaccinia virus. After synthesis of the GPCR DNA library, a Vaccinia virus library is created by Trimolecular Recombination[24] (left inset; yellow, regions homologous to the virus for recombination; blue, gene of interest). For this purpose, the randomized GPCR DNA library is first cloned into a *Vaccinia* virus transfer plasmid. The GPCR gene is then recombined with digested *Vaccinia* virus DNA, and infectious viral particles

are packaged using a fowlpox helper virus. The virus library is used to infect A-431 cells overnight at a multiplicity of infection of one virus per cell, covering the library diversity at a redundancy of 5–10. As the plasma membrane in mammalian cells is readily accessible from the extracellular space (ECS), expressed receptors can be directly labeled with a fluorescent ligand of choice. Cells that have higher receptor density will exhibit higher ligand binding and therefore higher fluorescence, which are then sorted by fluorescence-activated cell sorting (FACS). After sorting, the cells are lysed mechanically to release the virus, which is then amplified on fresh feeder cells. The sorted virus pool can then be used to infect for subsequent rounds of selection.

but not in microbial cells, on the selection outcome. For this purpose, R167[3.50] was mutated to Leu throughout the library, which was otherwise based on wild-type rNTR1. L167[3.50] disrupts the conserved D/ERY motif that is required for G protein coupling and thus fixing the receptor in an inactive state[14,27–29] (Supplementary Fig. 1).

The resulting cDNA library contained 2.9% mutations on average for each randomized position, resulting in 2.8 mutations per gene on average (Supplementary Fig. 1d, e). From this synthetic library, a viral library was created by Trimolecular Recombination yielding a diversity of $1.3 \times 10^8$, and A-431 cells were infected at an MOI of 1. As in previous microbial receptor evolution, selections were performed based on functional expression level. For this purpose, receptor-expressing cells were incubated with saturating concentrations of fluorescently labeled NT(8–13) and subjected to FACS whereby cells with the highest fluorescence levels (i.e. highest receptor expression) were enriched. After

each sorting round, virus was directly isolated from the enriched cell pools, amplified, and used to infect a fresh batch of cells for the subsequent selection round (Fig. 1b). In total, two selection rounds were performed by gating the top 0.5% and 0.06% of fluorescent cells, respectively. After each selection round, a right shift in the fluorescence signal of the sorted population was observed, indicating a higher receptor expression level in the selected pool (Fig. 2a). From each selection pool, NTR1 cDNA was amplified by PCR and subcloned into a mammalian expression vector and sequenced. 94 clones were isolated and screened for receptor expression levels, and the best expressing 25 clones were analyzed further. Overall, expression levels were between 25–82-fold of that of wild-type NTR1 and thus similar or even exceeding the levels that were reached by *E. coli*-evolved mutants[26]. In contrast, the underlying mutant NTR1-R167L only showed a 2.5-fold expression gain compared to wild-type NTR1, indicating that the

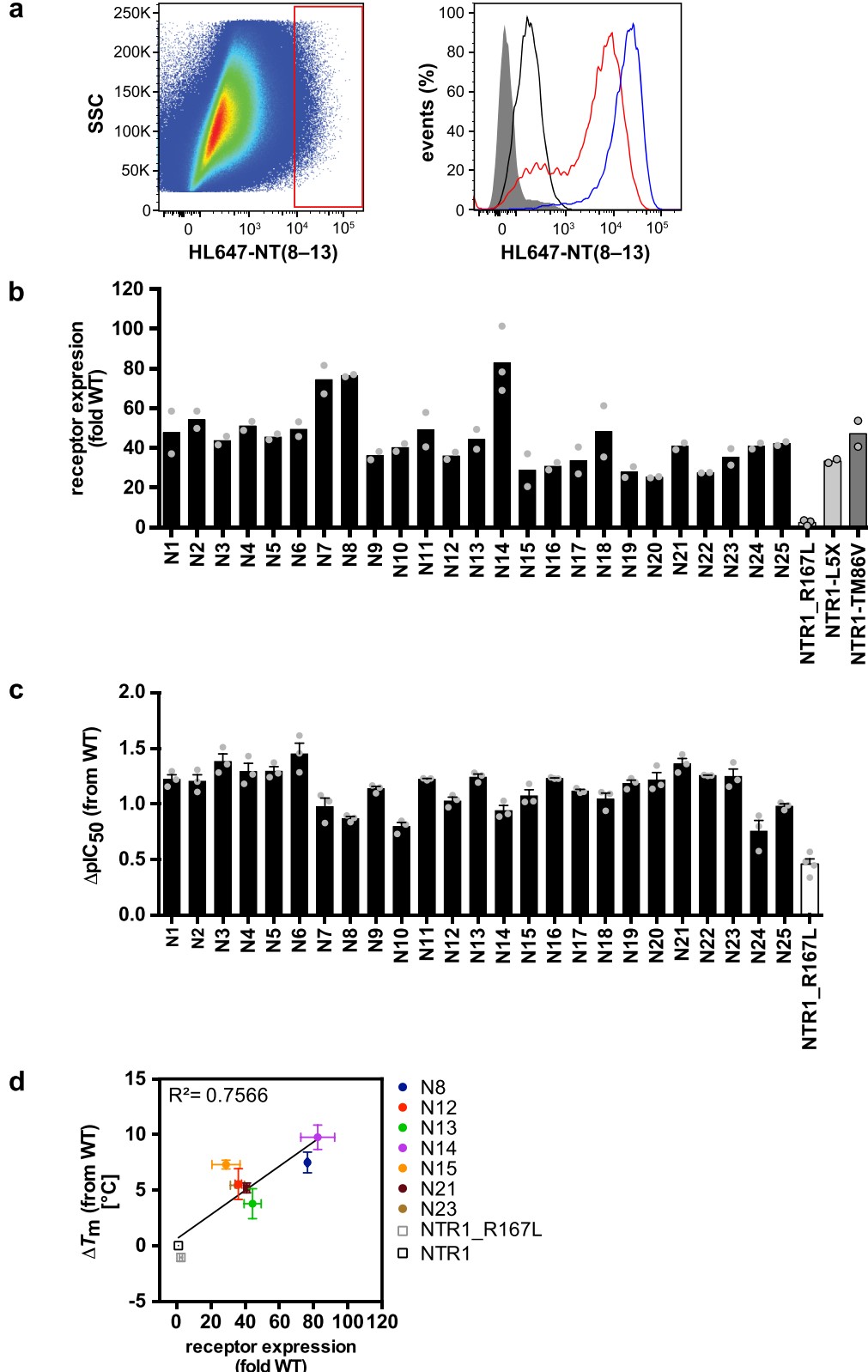

selection pressure was successful and most of the expression increase was accumulated during selections (Fig. 2b, Supplementary Table 1).

Like for the R167$^{3.50}$L mutant, signaling of all evolved NTR1 variants, all containing this mutation, was strongly impaired, reaching only 35% of the efficacy of wild-type NTR1 with the best mutants (Supplementary Fig. 2, Supplementary Table 1). Thus, the common R167$^{3.50}$L mutation had a strong impact on receptor function that

could not be overcome by evolution. The R167$^{3.50}$L mutation alone increased affinity for NT(8–13) ~3-fold, whereas evolved receptor variants exhibited up to 30-fold increased agonist affinity (Fig. 2c, Supplementary Table 1). This apparent discrepancy of achieving a higher affinity in the absence of direct mutations in the orthosteric binding site and in the absence of G protein coupling has been observed for several other NTR1 variants evolved in *E. coli* and some

**Fig. 2 | Evolution of NTR1 in mammalian cells yields receptor variants with improved biophysical properties. a** Sorting gates for improved NTR1 variants (left panel). The sorting gate for top 0.5% fluorescent cells is indicated as a red outline. Histogram comparing first and second sort populations post-amplification as compared to wild-type (right panel): sort 1 (red line), sort 2 (blue line), NTR1 (black line), negative control (dark gray shaded). **b** Expression analysis of 25 evolved NTR1 variants. Receptor expression was assessed in HEK293T cells by flow cytometry analysis with saturating concentrations of HL488-NT(8–13). Expression levels are relative to wild-type receptor expression and are given as mean values ± s.e.m. of 2 independent experiments. **c** Ligand affinities of 25 evolved NTR1 variants. IC$_{50}$ values were derived from whole-cell ligand competition-binding experiments with NT(8–13). Bars represent the mean change ± s.e.m. in calculated affinity (ΔpIC$_{50}$) for each mutant, compared with wild-type receptor from 3 independent experiments each performed in technical duplicates (Supplementary Table 1). **d** Correlation between receptor expression and thermostability. Thermostability of seven NTR1 variants was assessed in cell membrane fractions and is plotted as change in melting temperature (ΔT$_m$), measured by ligand binding, from wild-type receptor against expression levels from (**b**). Data represent mean values ± s.e.m. of 2–3 independent experiments performed in duplicates (Supplementary Table 1). Source data are provided as a Source Data file.

receptor mutants stabilized by other means as well[10,14,26,30]. It is likely a result of the disruption of the allosteric coupling between the agonist binding site and the intracellular G protein interface through disruption of the DRY microswitch and a stabilization of the receptor in an inactive conformation, despite exhibiting high-affinity binding[27,31]. This was further corroborated when analyzing the thermostability of several clones. All seven tested variants were strongly stabilized, and their melting temperatures ranged from 4–10 °C above wild-type NTR1, whereas the R167$^{3.50}$L mutation alone had no effect on thermostability (Supplementary Table 2). As in previous evolution campaigns performed in *E. coli*[26], also here a strong positive correlation between expression levels and thermostability was observed (Fig. 2d). Sequence analysis revealed between 5–8 mutations per variant. Three mutation hot-spots, A155$^{3.38}$T, Q239$^{5.36}$T and N365$^{7.49}$K, were present in 56%, 96% and 80% of the clones, respectively, suggesting a strong selection pressure on each of these residues (Supplementary Fig. 3). A155$^{3.38}$T and Q239$^{5.36}$T are not conserved in class A GPCRs and no clear correlation to increased thermo-stability can be made. The highly conserved N365$^{7.49}$ is part of the NPxxY microswitch and is coupled to the allosteric sodium site constituted by D$^{2.50}$ in class A GPCRs, which both are important mediators for signal propagation. Mutation of D$^{2.50}$ has been shown to effectively stabilize different class A GPCRs[8,32], yet no such mutation was found in our selection. Thus, it will be interesting to see whether the N365$^{7.49}$K mutation has similar structural and stabilizing effects as disruption of the sodium site.

Collectively, these data demonstrate that in mammalian cells highly stabilized and well expressing receptor variants can be obtained in only two rounds of selection that are comparable to the best expressing variants that were obtained by bacterial evolution. Similar to the variants obtained from bacterial selections where G protein coupling is not possible, the evolved receptors retaining the R167$^{3.50}$L mutation were strongly impaired in G protein signaling. This suggests that the evolved receptors were stabilized predominantly in inactive conformations, yet are still able to achieve high-affinity agonist binding, accounting for the improved biophysical properties. However, no sequence similarity was apparent between NTR1 variants evolved in mammalian cells and the *E. coli*-evolved variants NTR1-TM86V and NTR1-L5X. Thus, similar biophysical properties can be obtained through selection of distinct structural changes.

### Evolution of complex GPCRs in mammalian cells

GPCRs have evolved to sense a plethora of distinct stimuli at their extracellular side, which is reflected in great structural variability of the extracellular receptor regions. Some GPCRs contain extracellular domains (ECDs), which often exhibit complex folds that require the intricate quality control mechanisms of the mammalian cell for proper folding and translocation to the cell surface. Having demonstrated that with our system receptor stabilization and expression optimization can be achieved to extents that were similar or exceeded results from microbial systems, we were interested whether also receptors with more complex architectures would be amenable to selection in the mammalian system. To address this question, we chose parathyroid hormone 1 receptor (PTH1R) which belongs to the class B of GPCRs.

Receptors from this class structurally differ from class A GPCRs by an additional large extracellular domain that is required for binding peptide ligands. In addition, several posttranslational modifications of the ECD add significant structural complexity to the receptor fold[11,33–35], which hinders expression in microbes. Previously, we managed to evolve the TMD of PTH1R in yeast for higher expression and stability with an engineered small peptide ligand, which was a prerequisite for solving the crystal structure of the full-length receptor[11]. Yet, attempts to directly evolve the full-length PTH1R using the native 34 aa peptide ligand had failed, owing to toxic accumulation of misfolded receptor and due to the large size of the ligand, unable to reach the ligand-binding site in yeast cells despite permeabilizing the cell wall, demonstrating the limitations of microbial selection systems.

To perform evolution of PTH1R in the mammalian system, a synthetic library was created, following a semi-rational approach similar to the library for NTR1 described above. Randomization was restricted to the PTH1R TMD, thus avoiding changes to the ECD that would potentially affect ligand binding. As this work had been carried out before determination of the PTH1R structure[11], library design was based on a structural homology model of the human glucagon receptor. Thus, 118 positions within the TMD were selected for randomization, and here amino acid substituents were chosen based on chemical similarity. However, in contrast to the NTR1 library, no fixed mutations that would disrupt receptor function were introduced, as we were interested to investigate whether receptor properties beyond expression and stability could also be evolved in mammalian cells (Supplementary Fig. 4, c.f. Supplementary Notes). Nevertheless, several modifications were added to the library to be able to trace the selection results. We included 19 positions that had been identified in the yeast evolution campaign, of which 7 specific mutations kept the receptor in an inactive, highly thermo-stable conformation[11]. In contrast to the fixed R167$^{3.50}$L mutation in the NTR1 library, each of the yeast-derived positions was fully randomized, thus permitting selection of the wild-type or alternative residues at such a position during evolution. In addition, C351, which forms a disulfide bond between ECL (extracellular loop) 2 and the top of transmembrane helix 3[11], was randomized to test whether disruption of structurally relevant residues would corrupt selections (Supplementary Fig. 4C).

The resulting cDNA library contained 5.6% mutations on average for each randomized position, resulting in ~7 mutations per gene (Supplementary Fig. 4E–F). From this synthetic library, a viral library was generated yielding a diversity of $1.1 \times 10^8$, and A-431 cells were infected with the library at an MOI of 1. For the subsequent selections, two fluorescently labeled peptide ligands were used: first, the short peptide M-PTH(1–14) that had been engineered to only bind to the TMD of PTH1R, while still retaining the full agonistic potency of the native peptide[36], and second, the PTH analog PTH'(1–34) that resembles the native peptide agonist, as it requires interactions with both the ECD and the TMD of the receptor for high affinity binding[11,37,38]. In total, three selection rounds with each of the fluorescently labeled ligands were conducted, sorting the top 0.2–0.5% fluorescent cells

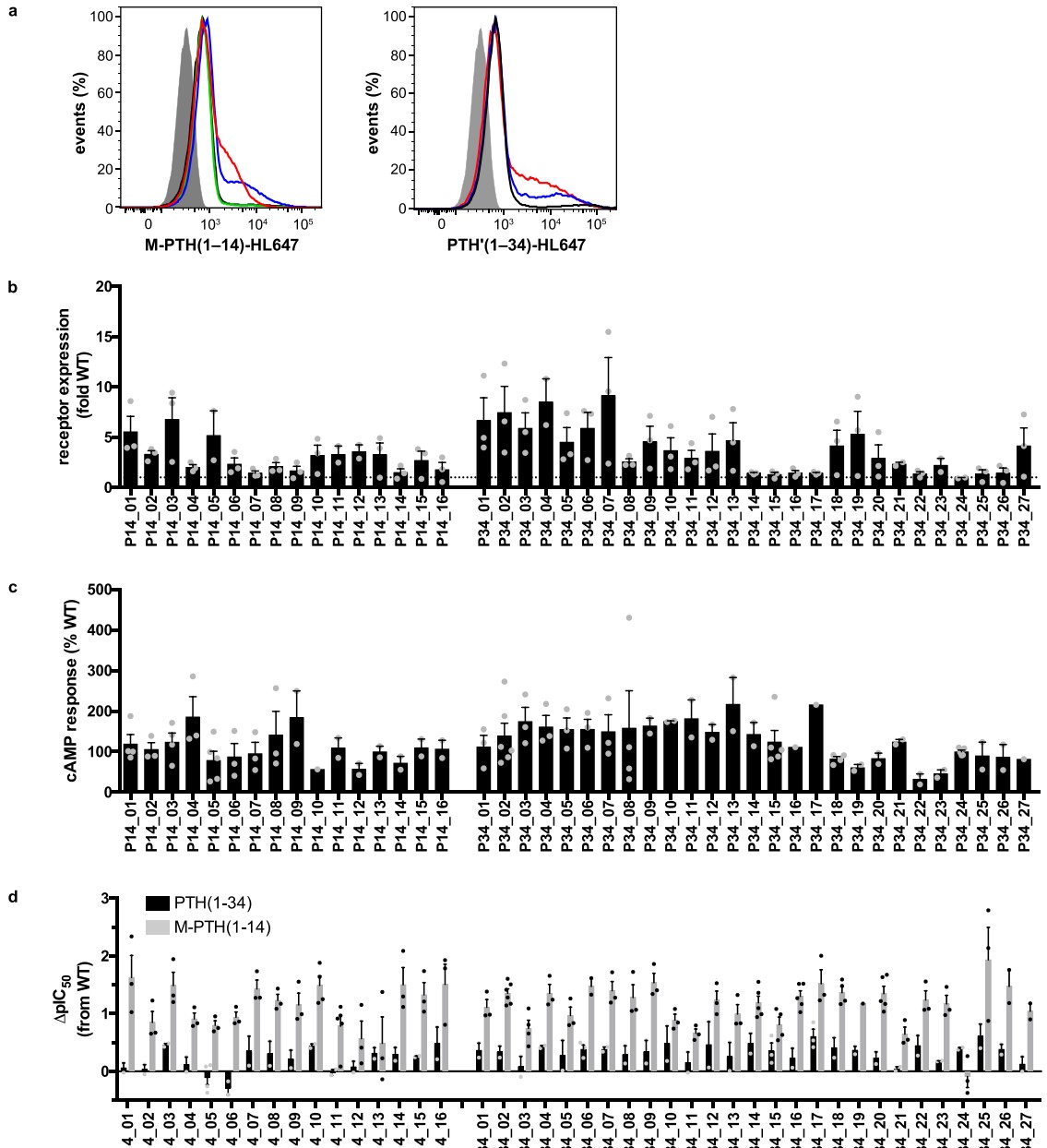

**Fig. 3 | Evolution of PTH1R in mammalian cells yields signaling-active receptor variants with increased ligand affinity. a** Comparison of the populations after selection with PTH'(1–34)-HL647 (left panel) or M-PTH(1–14)-HL647 (right panel) to wild-type PTH1R: sort 1 (red line), sort 2 (blue line), sort 3a (black line), sort 3b (green line, repetition of 3a), negative control (dark gray shaded). **b** Expression analysis of 43 evolved PTH1R variants assessed in live HEK293T cells by flow cytometry analysis with saturating concentrations of PTH'(1–34)-HL647. Expression levels are relative to wild-type receptor expression and are given as mean values ± s.e.m. of 2–3 independent experiments (Supplementary Table 3). **c** cAMP accumulation of 43 evolved PTH1R variants after stimulation with 1 μM PTH(1–34).

Data represent maximal cAMP concentrations relative to PTH1R. Bars represent mean values ± s.e.m. of 3–6 independent experiments performed in duplicates (Supplementary Table 4). **d** Ligand affinities of 43 evolved PTH1R variants in comparison to PTH1R. IC$_{50}$ values were derived from whole-cell ligand competition-binding experiments with M-PTH(1–14) or PTH(1–34). Bars represent the mean change ± s.e.m. in calculated affinity (ΔpIC$_{50}$) for each mutant compared with wild-type receptor from 2–8 independent experiments performed in duplicates (Supplementary Table 3). **b–d** Ligands used for selection are indicated below the bar plots. Source data are provided as a Source Data file.

(Supplementary Fig. 4G). Also here, higher cell fluorescence was observed with increasing selection rounds, albeit only a small population of highly fluorescent cells could be maintained (Fig. 3a). After the last round, 92 clones of each ligand selection were screened for expression, and the top expressing 43 variants of both pools were analyzed further. Like for the evolved NTR1 variants, an increase in expression was observed for all PTH1R variants, albeit to lower extents (up to 9-fold for PTH1R vs. up to 85-fold for NTR1) over wild-type (Fig. 3b, Supplementary Table 3). However, all PTH1R variants

remained signaling-active, with ~50% of the variants exceeding the maximal cAMP levels reached by wild-type PTH1R activation (Fig. 3c, Supplementary Table 4). This contrasts with the previous evolution of PTH1R in yeast, where highly stable but signaling-inactive variants were obtained[11]. Sequence analysis of the evolved PTH1R variants revealed a rather diverse set of mutations (Supplementary Fig. 5). In contrast to NTR1, no highly conserved mutations were found, and no conserved motifs of class B GPCRs were altered. Yet, L368[5.44], interacting with the conserved Val2[PTH/PTHrP] and M425[6.57] and T427[6.59] that interact with the

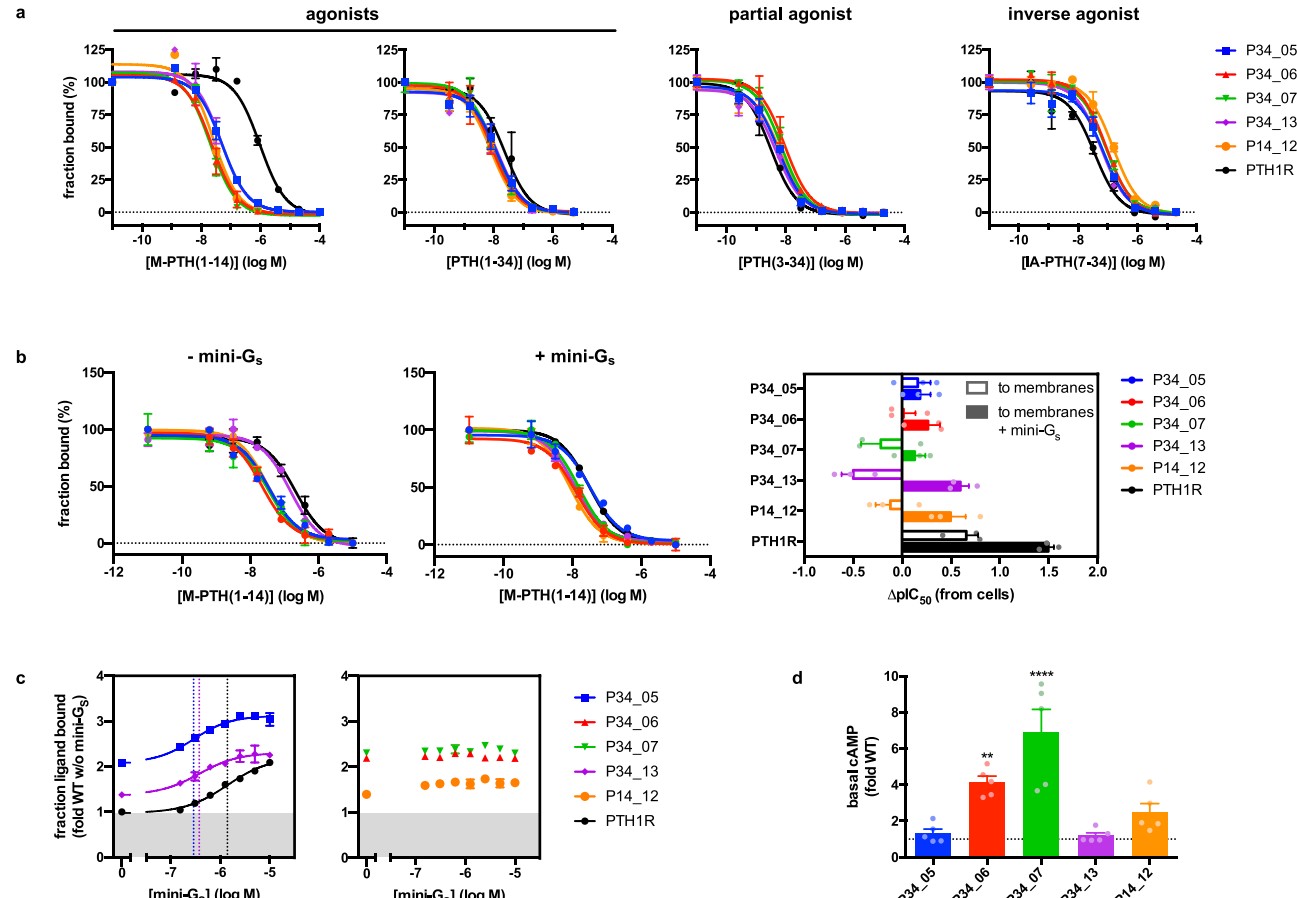

**Fig. 4 | Evolved PTH1R variants more easily adopt the active state. a** Evolved PTH1R variants exhibit high-affinity agonist binding but equal or reduced affinity for partial or inverse agonists in cells. Competition ligand binding curves of full [M-PTH(1–14) and PTH(1–34)], partial [PTH(3–34)] and inverse agonist [IA-PTH(7–34)], measured in whole cells expressing wild-type PTH1R or 5 evolved variants. **b** High agonist affinity of evolved PTH1R variants is similar to that of wild-type PTH1R in the G protein-bound state. Ligand binding curves of M-PTH(1–14) to evolved PTH1Rs were measured in membrane fractions, obtained from cells expressing PTH1R wild-type or 5 evolved variants, in the absence (left panel) or presence (middle panel) of 12.5 µM mini-$G_s$. Relative binding constants obtained by competition of M-PTH(1–14) binding with labeled M-PTH(1–14), measured in whole cells (from **a**) vs. membrane fractions in the absence or presence of 12.5 µM mini-$G_s$ (right panel). Positive values thus reflect stronger binding on membrane fractions than on cells. **c** Increased agonist affinity is G protein-dependent. Binding of 40 nM M-PTH(1–14)

was measured in membrane fractions supplemented with increasing concentrations of mini-$G_s$. Data are relative to binding levels of PTH1R in the absence of mini-$G_s$ (gray area). Left panel: Variants showing an increased basal and G protein-dependent increase in ligand binding. $EC_{50}$ values are indicated as dotted vertical lines (PTH1R: 1.39 ± 0.2 µM, P34_05: 0.28 ± 0.06 µM, P34_13: 0.38 ± 0.1 µM). Right panel: Variants showing an increase in ligand binding independent of G protein. **d** G protein-independent increase in ligand binding is due to higher basal activity. cAMP levels were measured 30 min after addition of the phosphodiesterase inhibitor IBMX to cells expressing PTH1R variants and were normalized to receptor expression levels. Data represent mean values ± s.e.m. of 3 experiments performed in duplicates. **p = 0.0044, ****p < 0.0001. Statistical significance was determined by one-way ANOVA and Bonferroni multi-comparison. Source data are provided as a Source Data file.

N-terminal residue of the PTH peptides in active-state PTH1R[38–40] were found to be mutated in more than 25% of the selected variants. Thus, these mutations could provide a mechanistic rationale for the observed increase in receptor potency and require further analyses.

As we had allowed the same mutations in the present library as in the yeast library, it was interesting to observe that those mutations which contributed to receptor stabilization but at the same time disrupted receptor signaling, were deselected in the mammalian system, and that in almost all positions the wild-type variant was preferred (Supplementary Fig. 6). Thus, mutations that had been critical for the success of stabilization of an inactive receptor conformation in yeast cells were not preferred in a signaling competent environment. Likewise, mutations of C351 were completely rejected, indicating that formation of the disulfide between ECL 2 and helix 3 is an important trait during selection (Supplementary Fig. 6), which is in line with its known relevance in receptor function[33]. Also, with respect to ligand affinity, distinct differences to the previously evolved NTR1 variants were observed. Binding affinity of PTH(1–34) was mostly unaltered or

only modestly (0.6–5-fold) increased for the PTH1R variants. However, most receptor variants exhibited strongly increased affinity for M-PTH(1–14), indicating a fundamental difference in the binding of both ligands, where PTH(1–34) can rely on its binding to the ECD (Fig. 3d, Supplementary Table 3). However, these differences seemed not to be induced by the type of ligand during the selection as no differences with respect to the apparent receptor properties between variants from either selection regime were detected. Taken together, in mammalian cells it was possible to evolve full-length PTH1R with a set of two ligands differing in size, which had not been possible in microbial systems due to limitations imposed by the expression host. Moreover, receptor variants were obtained that remained signaling-competent, demonstrating the importance of the cellular context for the selection outcome.

## Tuning allostery of a class B GPCR by directed evolution
For NTR1, uncoupling of the allosteric transmission from the binding pocket to the G protein interface likely allows for the accumulation of

increased agonist affinity, thus directing the receptor into an agonist-binding, inactive but stable conformation. This was not the case for evolved PTH1R. When we assessed the thermostability of five evolved receptor variants in isolated membrane fractions, all tested variants exhibited decreased thermostability compared to wild-type PTH1R (Supplementary Fig. 7A, Supplementary Table 5). Given the retained signaling ability in combination with increased ligand affinity of the evolved PTH1R variants and the fact that directed evolution was carried out in mammalian cells expressing G proteins, we speculated that the selected mutations may have optimized the receptor for this context. Consistent with this idea, in the presence of mini-$G_s$, thermostability increased for most mutants (Supplementary Fig. 7B), and for P34_05 the decrease was diminished, giving a relative improvement for all mutants (Supplementary Fig. 7C). This became even more apparent when we compared the binding of different ligands in whole cells. As shown above, affinities of both agonists, M-PTH(1–14) and PTH(1–34), were increased for the PTH1R variants when compared to wild-type PTH1R (Figs. 3d, 4a). However, for the partial agonist PTH(3–34), and more pronounced for the inverse agonist IA-PTH(7–34), a reversal in relative affinities was observed, where wild-type PTH1R had a slightly higher apparent affinity than the evolved variants (Fig. 4a). According to the ternary complex model of GPCR signaling, G protein association shifts the conformational equilibrium of the receptor towards an active state, which allosterically increases agonist affinity and in turn decreases antagonist affinity[41–43]. We thus speculated that the evolved PTH1R variants may have incorporated mutations that stabilize a conformation favoring G protein binding, thus shifting the receptor equilibrium towards a high-affinity state for agonists in the presence of G proteins.

To test this hypothesis, we directly compared the effect of G protein on ligand binding. Affinity of M-PTH(1–14) to wild-type PTH1R in isolated membrane fractions was increased 5-fold compared to its affinity in whole cells, which likely reflects the formation of a more stable receptor G protein complex due to lower nucleotide concentrations in membrane fractions[42]. In contrast, agonist affinities of the evolved variants remained identical to their affinities in whole cells. Only for variant P34_13, a decrease in affinity to the level of wild-type PTH1R by going from cells to membranes was observed. By adding mini-$G_s$ to the membrane fractions, which mimics the activated state of the G protein[44], wild-type PTH1R affinity was further shifted towards that of the evolved variants, suggesting that indeed the high-affinity state of the evolved variants was due to higher propensity to adopt an active-state conformation (Fig. 4b).

To corroborate these findings, we further assessed the influence of G protein to induce a high-affinity state for each of the variants. For this purpose, receptor occupancy by M-PTH(1–14) at submaximal concentrations was measured as a function of increasing G protein concentrations (Fig. 4c). For all evolved receptor variants already in the absence of externally added G protein, a larger fraction of receptor was ligand-bound when compared to wild-type PTH1R, reflecting the apparent high-affinity state observed in the competition binding curves. As expected, with increasing G protein concentration, the fractional binding of M-PTH(1–14) to wild-type PTH1R was shifted up to 2-fold, indicating the transition of the receptor into a high-affinity state induced by G protein binding. Variants P34_05 and P34_13 had a similar behavior, as both variants were shifted into a high-affinity state with increasing added G protein concentrations. However, 4–5-fold lower $EC_{50}$ values suggested that both receptor variants exhibited a higher affinity for G protein, thus reaching the high-affinity state more easily. In contrast, variants P34_06, P34_07 and P14_12 were already in a high-affinity state, which was independent of added G protein. Likewise, the latter variants exhibited increased basal receptor activity, whereas basal activity of variants P34_05 and P34_13 was similar to that of wild-type PTH1R (Fig. 4d). Therefore, evolution of PTH1R with an agonist in a signaling-competent environment has led to the accumulation of mutations that favor transition of the receptor into an active G protein-bound state or that promote basal receptor signaling being in a conformation compatible with agonist binding, which both result in a higher apparent affinity for agonists.

## Discussion

Bacterial and yeast selection systems have been used for directed evolution of GPCRs since large libraries can be created, where each cell carries a single receptor variant. The high efficiency of transformation and the establishment of a rather uniform plasmid copy number ascertain that the linkage between genotype and phenotype is preserved and both can be readily determined. However, applicability of microbial systems is limited to relatively small ligands that can be used for selection because of the outer barrier of a cell wall or an outer membrane. Furthermore, since minimal expression of the native receptor at the beginning of the selection process is required, toxicity of some receptors can be a problem.

Here, we demonstrate the development of a system that has no such limitations. Mammalian cells offer a native cellular environment for GPCRs, including quality control during export, membrane composition and posttranslational modification machinery, resulting in optimal expression conditions even for complex receptors. At the same time, the ligand binding pocket is readily accessible at the cell surface for ligands of any size and composition. Even conformation-specific modulators binding to the extracellular surface such as antibody fragments could be used as selection tools. Moreover, in mammalian cells, GPCRs are integrated into the endogenous cellular signaling framework. This offers the possibility to even interrogate signaling pathways and thus direct selection towards specific signaling properties.

However, creation of clonal libraries in mammalian cells is difficult. Standard transfection methods are not suitable because they are relatively inefficient and typically several hundred plasmids − with a considerable spread from cell to cell − are taken up by each cell, precluding the coupling of genotype and phenotype by any means. Key to our development was leveraging a highly efficient viral transduction system which is based on a *Vaccinia* vector[24]. It combines the possibility to tune transduction rate to an average of one particle per cell, thus maintaining monoclonality, coupled with an inherent vector amplification reaching rather uniform copy numbers. Another feature is a unique recombination module in the viral vector that enables the creation of highly diverse libraries which are also compatible for large proteins such as GPCRs.

While commonly random mutagenesis is being used to generate genetic diversity, we have instead created synthetic GPCR libraries by codon-based solid-phase synthesis[45,46], which gives full control over position and composition of randomization, as well as avoiding frameshifts and stop codons. Thus, prior structural and functional knowledge can be incorporated into the design to improve selection outcomes, to exclude unwanted selection events a priori and to introduce predefined bias to the library, e.g., for amino acid types or a percentage of wild-type codons. Moreover, here we used this approach to test the fidelity of the selection system by spiking the libraries with specific mutations with predictable effects. For NTR1, the crystal structure[10] served as a template to identify suitable residues for randomization, and we restricted mutagenesis to the most common corresponding amino acids from all class A GPCRs, thus maintaining an evolutionary context and excluding potentially detrimental mutations. For PTH1R, we designed the library based on a structural homology model and chemically conservative mutagenesis, as neither structural information nor sufficient phylogenetic data had been available for PTH1R at the time. This demonstrates that even in the absence of accurate structural information, potent libraries can be generated. In the future, even more sophisticated library designs may be derived

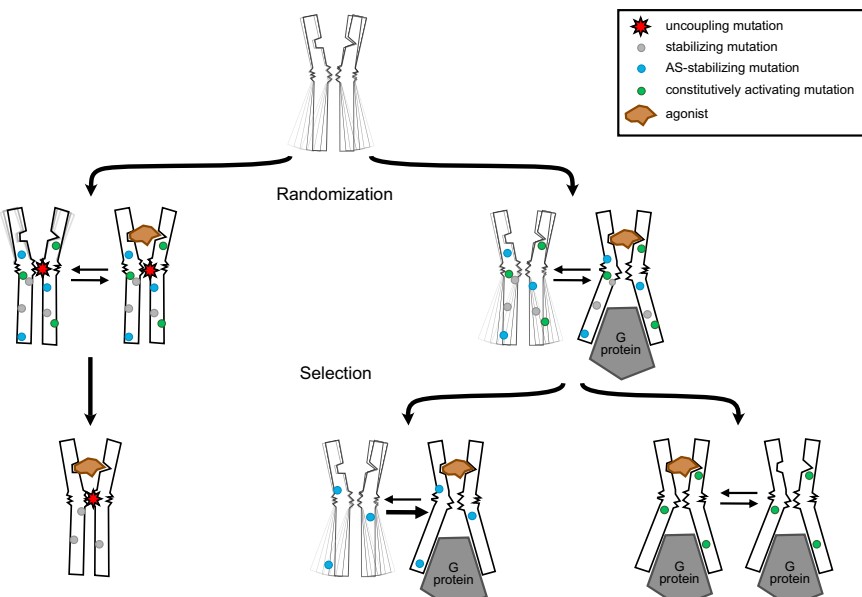

**Fig. 5 | Modulation of biophysical and allosteric properties of GPCRs by directed evolution in mammalian cells.** GPCRs sample a multitude of conformational states that are allosterically modulated by ligand and G protein binding. Introduction of an uncoupling mutation (red star) leads to disruption of the signal transmission from the ligand binding pocket to the G protein interface (left path). Thus, in subsequent directed evolution in the absence of G protein or when its binding was prevented by a mutation already stabilizing the inactive state, preferentially mutations that further stabilize the receptor in an inactive state (grey) were selected, leading to a rigid and stable conformation. If receptor function is retained at the beginning of selection, the cellular environment, which contains G proteins, dictates the selection outcome. Mutations promoting allosteric coupling between the agonist-occupied conformation of the ligand binding pocket and the active-state (AS) conformation of the G protein binding interface (blue) were selected. In some cases, mutations were enriched that promote a strong allosteric coupling of the active-state conformation, resulting in constitutive receptor activity (green). Figure modified after Nygaard et al.[60].

from the steadily increasing amount of structural and functional information on GPCRs that can be easily incorporated into the workflow.

Our first goal was to recapitulate microbial selections, i.e., receptor evolution in the absence of any downstream effectors such as G proteins, in order to compare the fidelity of this mammalian system to previous approaches. Therefore, the library for NTR1 was designed to contain the R167[3.50]L mutation, which disrupts the DRY motif in the receptor core, required for G protein coupling[14,27–29]. Notably, while the R167[3.50]L mutation by itself had no effect on stability and only a modest effect on expression levels compared to wild-type NTR1, it allowed uncoupling of the receptor from the G protein, which led to the accumulation of mutations during the selection that simultaneously and strongly increased expression levels and thermostability (Fig. 5). Several variants even exceeded expression levels of receptors that had undergone extensive expression-guided evolution in *E. coli*[26]. Also, in line with previous selections in microbial systems, all evolved variants exhibited increased agonist affinity, despite their inability to activate G proteins. Importantly, this fact did not preclude selection for the desired features, higher functional expression and protein stability. As observed in structures of previously stabilized receptors[10,14], this may suggest that uncoupling the signal transmission path had two consequences on the selection outcome: first, mutations were enriched that rigidified the extracellular part of the receptor commensurate with agonist binding, with the consequence of promoting of high-affinity agonist binding, thus again contributing to receptor stability. Second, the intracellular part of the receptor was stabilized through mutations keeping the receptor in an inactive conformation, favored by the presence of R167[3.50]L (Fig. 5). In short, the two parts of the receptor may be viewed as having been independently evolved.

In contrast, the PTH1R library contained no fixed mutations that would be expected to interfere with downstream signaling. As a consequence, endogenous G proteins were in principle able to interact with the receptor and to influence the selection outcome. Indeed, mutations that had been deliberately included into the library and that would stabilize the receptor at the cost of a reduced signaling capacity were not found under these selection conditions. In line with that finding, selected receptor variants exhibited even lower thermostability compared to wild-type PTH1R in membrane preparations devoid of G proteins, suggesting that a fundamentally different pathway for selection had been followed. The expected increased stability was seen as soon as G protein was added to the membranes.

Like for NTR1, a general increase in receptor expression and in ligand affinity was observed for most variants, independent of the peptide agonist that had been used for selection. The gain in ligand affinity was more pronounced for M-PTH(1–14) than for native PTH(1–34). Considering that mutations were restricted to the TMD and that M-PTH(1–14) makes no additional contacts to the ECD, it was reasonable to assume that changes within the TMD were accounting for increased ligand affinity. The less pronounced differences in PTH(1–34) affinity can be explained by the additional energy that is provided by interactions of the C-terminal part of the peptide binding with the ECD in class B GPCRs[47].

Interestingly, the apparent increase in agonist affinity for the evolved PTH1R variants was most prominent in whole cells. Agonist binding to a GPCR shifts its conformational equilibrium to a state that favors interaction with G proteins[48,49] and thus leads to activation of the G protein by destabilization of the nucleotide-binding pocket and GDP dissociation[50]. The resulting nucleotide-free ternary complex often exhibits increased agonist binding affinity[41], but it is extremely short-lived in vivo due to high intracellular GTP concentrations, resulting in rapid binding of GTP to Gα[51]. In membrane fractions, which are nucleotide-depleted in comparison to whole cells, a relative increase in agonist affinity of wild-type PTHR was observed, reflecting the higher stability of ternary complexes in the absence of nucleotides. This was even more pronounced when supplementing membrane fractions with mini-Gs, mimicking the nucleotide-free state of Gα.

Under these conditions, wild-type receptor exhibited agonist affinity comparable to that of the evolved variants.

Thus, during the selections of PTH1R, mutations were acquired which allow the receptor to transition into a high-affinity state even without the help of an active-state G protein conformation. Hence, the allosteric coupling of the ligand binding site with the active but not the inactive conformation of the G protein binding site is enhanced by the selected mutations, resulting simultaneously in higher agonist and G protein affinity in many of the selected mutants[31] (Fig. 5). In line with this finding, the apparent affinity for mini-$G_s$ was increased for two of the evolved variants. Notably, for three variants we could not observe any additional influence of mini-$G_s$ on agonist occupancy of the receptor. Intriguingly, those three variants showed increased constitutive receptor activity, which explains why the receptor high-affinity state is not further altered by G protein addition.

In the present study, we used ligand binding as a surrogate for expression of correctly folded and integrated receptor and as primary selection pressure. While this has yielded expression- and stability-optimized receptor variants for NTR1, this was not the case for PTH1R when measured as for NTR1. PTH1R was permitted to evolve in a fully functional signaling environment, and we observed that the selection was strongly impacted by receptor function. Even without applying direct selection pressure on G protein binding, we obtained receptor variants that did achieve higher stability, but only when allowed to engage a G protein. Yet, unlike in the NTR1 selections, during PTH1R selections, it was more difficult to maintain the pool of cells with high fluorescent levels. We cannot exclude that under the stress of the FACS selection with a high-copy viral vector, some high-expressing clones are lost, while others are compatible. Moreover, variants exhibiting high constitutive activity may have been lost during the course of selection due to constant internalization rendering the receptor inaccessible for the fluorescent ligand and possibly also impairing cell viability due to high basal signaling.

Taken together, this mammalian selection system has allowed us to select for receptors exhibiting strongly altered functional properties that will be valuable tools to study the mechanisms of signal transmission in this receptor class. The stabilized receptors can be useful in fragment-based drug discovery, where the initial low potencies preclude functional assays, and hit-finding has to rely on binding assays, e.g. by NMR and SPR[16] on solubilized receptors. Mutants with high constitutive activity may be especially interesting candidates for drug screening approaches[52]. As noted above, some of these mutants might be missed with the current setup, however a way out could be to use other selection parameters that directly measure downstream signaling. In recent years, a multitude of sensor systems have become available to study GPCR function at all levels of the signaling cascade in mammalian cells. Many of these assays are fluorescence-based and are in principle compatible with flow cytometry applications. Such assays include receptor-G protein, receptor-arrestin or even G protein interaction assays which are based on split fluorophore systems or FRET sensors[53–57]. Integration of such sensors for receptor selection will further increase the versatility of our mammalian evolution system, as receptor evolution can be directed towards specific functional properties. This strategy should prove particularly useful for deciphering the mechanisms of signaling bias and regulatory mechanisms of GPCR-mediated signal transduction. Moreover, it enables the repurposing of membrane receptors as biosensors for drug screening applications and could be used to create tools for optogenetic applications.

## Methods

### Ligands

Human PTH(1–34) and PTH(3–34) were from Bachem. Neurotensin 8–13 [NT(8–13)] was from Anaspec. Human [Ac5c$^1$, Aib$^3$, Q$^{10}$, Har$^{11}$, A$^{12}$, W$^{14}$]PTH(1–14), [M-PTH(1–14)], was synthesized by Peptide Specialty Laboratories. M-PTH(1–14) was labeled at K$^{13}$ with HiLyte dye 647 [M-PTH(1–14)-HL647]. [Nle$^{8,18}$,Y$^{34}$,C$^{35}$]PTH(1–34) was labeled at C$^{35}$ with HiLyte dye 647 [PTH'(1–34)-HL647, the prime indicating the sequence changes compared to PTH(1–34)]. Neurotensin 8–13 was fluorescently labeled with HiLyte-647 [HL647-NT(8–13)] or HiLyte-488 [HL488-NT(8–13)] at the N-terminal amino group. All fluorescently labeled peptides were custom synthesized by Anaspec.

### Cell culture

HEK293T cells (Cat. No. CRL-11268), A-431 (Cat. No. CRL-1555) and BS-C-1 cells (Cat. No. CCL-26) were from ATCC and were cultivated in Dulbecco's modified medium supplemented with 10% (v/v) fetal calf serum. Cells were maintained at 37 °C in a humidified atmosphere of 5% $CO_2$, 95% air. Transient transfection of HEK293T cells was performed with TransIT-293 (Mirus) reagent according to the manufacturer's protocol. CHO-S cells (Life Technologies, Cat. No. R80007) were maintained as shaking suspension culture using Power CHO 2CD medium (Lonza) supplemented with 8 mM L-glutamine, 0.1 mM hypoxanthine and 0.1 mM thymidine. Cells were seeded in DMEM with 10% fetal calf serum overnight to facilitate adhesion before transfection. Transient transfection was performed with Lipofectamine reagent (Invitrogen) according to the manufacturer's protocol.

### DNA library design and synthesis

cDNA libraries for rat NTR1 and human PTH1R were custom-synthesized by MorphoSys AG as linear DNA fragments using the Slonomics® technology[26,45,46]. Constant parts of the sequence were amplified from a wild-type-containing plasmid. To generate the variable parts, mixtures of anchor molecules were generated in a defined ratio to represent all combinations of two adjacent positions within a variable region[26,45,46]. These mixtures were connected to the growing DNA chain by ligation. The reaction product was purified by immobilization on a streptavidin-coated surface (Microcoat). New overhangs for the next reaction cycle were then generated by restriction with the enzyme Eam1104I (Thermo Scientific), and a new mixture of anchor molecules was added. After three to seven reaction cycles, product pairs were combined to generate transposition intermediates. These were either combined in a second round to form long variable regions or assembled with constant parts by restriction and ligation. Length variants were synthesized separately, quantified by PAGE, mixed in a defined ratio, and assembled as one pool. The PTH1R library was synthesized separately in two equally sized fragments and combined by restriction with Esp3I (Thermo Scientific) in the flanking region and subsequent ligation. The final ligation product was amplified to ~2 μg by PCR using Phusion DNA polymerase (NEB).

### NTR1 DNA library cloning

Acceptor constructs for rat NTR1, containing a cloning cassette with the *Mus musculus* IgG signal sequence aa 1–17, were constructed for both the mammalian expression plasmid (EFMOD, Vaccinex, 5' BssHII and 3' SalI cloning sites) and *Vaccinia* transfer plasmid (VHEH5, Vaccinex, 5' BssHII and 3' BsiWI cloning sites). The wild-type rat NTR1 gene (amino acids 43–424), along with NTR1 variants, L5X and TM86V, were amplified using PCR (iProof high-fidelity DNA polymerase, BioRad) and the following primers: NTRBSSHIIsense 5'-ttttt*GCGCGC*ACTCCAC CTCGGAATCCGACACGG-3' and NTRaddSal1 5'-tttt*GTCGAC*TCAGTAC AGGGTCTCCCGGGGTG-3' (for EFMOD) or NTRAddBsiW1stop-5'-ttttt*CGTACG*tTCAGTACAGGGTCTC- 3' (for VHEH5) by standard protocols. PCR products were subsequently cloned into the expression and transfer plasmids. DNA library construction was performed by PCR (Advantage2 polymerase, Clontech) of the mutant library 2218_ −1_LIB_rNTR1 (43-424) using the same primers. The PCR product was resolved on 1% agarose/TBE gels. The 1177 bp band was gel-purified (Qiaquick, Qiagen), digested and ligated into the VHEH5 plasmid using NxGenT4 DNA ligase (Lucigen). High-efficiency transformation was

done by electroporation of NEB10Beta *E. coli* cells (BioRad GenePulser, 1 mm cuvette, 2.0 kV, 200 Ω, 25 μF) to create a plasmid library with a diversity of ~1.4 × 10⁷.

## PTH1R DNA library DNA cloning

Acceptor constructs for human PTH1R with a cloning cassette containing the PTH1R signal sequence aa 1–23 (including a naturally occurring BsiWI site) and a 3′ SalI site were constructed for both mammalian expression (EFMOD, Vaccinex) and the inducible *Vaccinia* transfer plasmid (T7terVHE, Vaccinex). The full-length wild-type human PTH1R gene (1–593) were amplified using PCR (Q5 DNA polymerase, NEB) and cloned into expression and transfer plasmids (BsiWI/SalI). DNA library construction was performed by PCR (Q5 DNA Polymerase, NEB) of the linear DNA of mutant library SLN2248 with standard conditions and minimal cycling using the following primers: PTH1Rsignalsense 5-CTCAGCTCCG<u>CGTACG</u>CGCTGGTG-3′ and PTH1R AS 5′-TGTCCGTTCG<u>GTCGAC</u>TCACATGACTGTCTCC-3′. The PCR product was resolved on 1% agarose/TBE gels. The 1734 bp band was gel purified (Qiaquick, Qiagen), digested with BsiWI/ SalI and ligated into T7TerVHE plasmid using NxGenT4 DNA ligase (Lucigen). High-efficiency transformation was described above to create a plasmid library with a diversity of ~6.3 × 10⁶.

## Virus generation by trimolecular recombination

*Vaccinia* vector V7.5 virus (Vaccinex) was digested with Proteinase K (Thermo Fisher), and DNA was purified by phenol/chloroform extraction. V7.5 viral DNA was digested with restriction endonucleases ApaI (NEB) and NotI (NEB) and purified with Amicon ultra centrifugal columns (Millipore Sigma). BSC-1 cells were infected with helper fowlpox virus at a multiplicity of infection (MOI) of 1.5 plaque-forming units (pfu) per cell and transfected with digested V7.5 vector DNA and each receptor library and the corresponding control plasmids. Infected/transfected cells were incubated for 5 days, and *Vaccinia* virus was harvested by freeze-thawing the cells. Individual plaques for control clones were picked and amplified. Viral DNA was purified and amplified by PCR. Positive clones were confirmed by sequencing. Virus stock for the library was titered by plaque assay. Individual clones were randomly picked, and checked by PCR for recombination efficiency. The resulting *Vaccinia* libraries had over 95% positive recombinant efficiency harboring ~1.3 × 10⁸ and ~1.1 × 10⁸ unique recombinants for NTR1 and PTH1R, respectively.

## *Vaccinia* virus infection and fluorescent ligand binding

A-431 cells were seeded the day before infection in DMEM + 10% (v/v) FBS and allowed to double overnight at 37 °C, 5% CO₂. Cells were then infected overnight at an MOI of 1 pfu per cell with virus expressing NTR1 controls or the library. The appropriate pfu of virus was diluted into a minimal medium volume to cover the cell monolayer and incubated at 37 °C, 5% CO₂ for 1–2 h. Cells were then overlaid with sufficient media and allowed to incubate for 16–18 h. Cells were harvested using Accutase™, pelleted and washed in FACS buffer [PBS, 1% (w/v) BSA] or Tris buffer [20 mM Tris-HCl (pH 7.4), 118 mM NaCl, 5.6 mM glucose, 1.2 mM KH₂PO₄, 1.2 mM MgSO₄, 4.7 mM KCl, 1.8 mM CaCl, 0.1% (w/v) BSA]. Cells were resuspended at 2 × 10⁶ cells per ml in FACS buffer or Tris buffer and incubated with fluorescent ligand on ice for 1–2 h. To confirm specificity, duplicate cell samples were also incubated with a 100-fold excess of unlabeled ligand. Cells were then washed in the appropriate buffer and fixed in 0.5% paraformaldehyde with propidium iodide for live/ dead cell discrimination before analysis on the flow cytometer. The gating strategy is exemplified in Supplementary Fig. 8.

## Fluorescence-activated cell sorting of improved NTR1 variants

A-431 cells, infected with a library of NTR1 clones, were sorted for multiple iterations using 40 nM NT(8–13)-HL647. For the first round of sorting, 4 × 10⁷ A-431 cells were infected with the NTR1 library at an MOI of 1 overnight at 37 °C, 5% CO₂, as described above. The next day, the cells were harvested using Accutase™ and stained with 40 nM of the ligand in 1 ml total volume FACS buffer on ice for one hour with occasional, gently swirling. Cells were then washed twice with FACS buffer, resuspended at 2 × 10⁷ cells per ml and passed through a 40 μm filter before being sorted on the BD FACS Aria sorter. The top 0.3% fluorescent cells were collected (6800 total), lysed by multiple freeze/thaw cycles and the virus was amplified in multiple flasks of BSC-1 cells for 2–3 days.

The amplified virus was harvested and titered before being used to infect A-431 cells again for a second round of sorting. Since the diversity of the pool from the first sort was only 6800, 3 × 10⁶ A-431 cells were infected for the second sort. The top 0.06% events were collected and amplified as above. Enrichment in the sort was tested by small-scale infections and ligand staining throughout.

## Fluorescence-activated cell sorting of improved PTH1R variants

A-431 cells infected with a library of PTH1R clones were sorted for multiple iterations using 120 nM M-PTH(1–14)-HL647 or 120 nM PTH′(1–34)-HL647. For the first round of sorting, 1.2 × 10⁸ A-431 cells were infected with the PTH1R T7-inducible library and the attenuated T7 promoter virus at an MOI of 1 overnight at 37 °C, 5% CO₂, as described above. The next day, cells were harvested using Accutase™ and stained with 120 nM of ligand in 6 ml total volume on ice for one hour with occasional, gently swirling. Cells were then washed twice with FACS buffer, resuspended at 2 × 10⁶ cells per ml and passed through a 40 μm filter before being sorted on a BD FACS Aria sorter. The top 0.5% fluorescent cells were collected, lysed by multiple freeze/thaw cycles, and the virus was amplified in multiple flasks of BSC-1 cells for 2–3 days.

The amplified virus was harvested and titered before being used to infect A-431 cells again for a second round of sorting. Each subsequent round of sorting was performed with 1.5 × 10⁷ A-431 cells and gating stringency was increased for each round. Sort enrichment was tested as small-scale infections and ligand staining throughout.

## Isolation of receptor variant DNA and cloning into mammalian expression vectors

*Vaccinia* DNA was extracted from the sorted pools (DNA Blood mini, Qiagen). Pool variants were amplified from the pool DNA using PCR (Advantage2 polymerase, Clontech) by standard protocols with minimal cycling. For NTR1, primers NTRBSSHIIsense 5′-ttttt<u>GCGCGC</u>ACTCCACCTCGGAATCCGACACGG-3′ and NTRaddSalI 5′-tttt<u>GTCGAC</u>TCAGTACAGGGTCTCCCGGGTG-3′ (EFMOD), and for PTHR1, signal sense 5′-CTCAGCTCCGCGTACGCGCTGGTG-3′ and PTHR1AS 5′-CCCCCCTCGAGGTCGACTCACATGACTGTCTCCC-3′. PCR products were subsequently cloned into the mammalian expression vector EFMOD. Mini-libraries were prepared by picking 92–94 colonies and isolating plasmid DNA (Qiaprep 96 turbo, Qiagen). DNA sequences were analyzed by Sanger sequencing using 2–3 primers for full coverage.

## Preparation of mini-G$_s$ protein

Mini-G$_s$ 393 was essentially prepared as described before[44]. Briefly, mini-G$_s$ was expressed in *E. coli* strain BL21(DE3) at 20 °C. Cells were harvested 16–20 h post-induction by centrifugation, resuspended in lysis buffer [40 mM HEPES (pH 7.5), 150 mM NaCl, 5 mM imidazole, 10% (v/v) glycerol, 5 mM MgCl₂, 50 μM GDP, 1 mM DTT, 50 μg/ml DNAseI, 50 μg/ml lysozyme] and disrupted in a HPL6 cell lyser (Maximator GmbH) at 1700 bar. Lysates were clarified by centrifugation (20,000 *g* for 45 min) and supernatants were loaded on Ni-NTA columns (Thermo Fisher Scientific). Columns were washed with 10 CV wash buffer [20 mM HEPES (pH 7.5), 500 mM NaCl, 28 mM imidazole, 10% (v/v) glycerol, 1 mM MgCl₂, 50 μM GDP, 1 mM DTT], and bound proteins were eluted stepwise in 2 CV of elution buffer [20 mM HEPES (pH 7.5),

150 mM NaCl, 500 mM imidazole, 10% (v/v) glycerol, 1 mM MgCl$_2$, 50 μM GDP, 0.5 mM DTT]. Imidazole was removed on PD-10 desalting columns (Cytiva), proteins were concentrated to 25 mg/ml and snap-frozen in freezing buffer [25 mM HEPES (pH 7.5), 150 mM NaCl, 15% (v/v) glycerol, 5 mM MgCl$_2$, 10 μM GDP, 0.25 mM DTT].

## Expression analysis

Receptor variants were transiently transfected in HEK293T cells. 48 hrs after transfection, cells were detached with Accutase™ and incubated with 20 nM HL488-NT(8–13) or 100 nM PTH'(1–34)-HL647 in PBS supplemented with 0.2% BSA for 2–4 h on ice. Nonspecific binding was determined in the presence of a 100-fold excess of unlabeled peptide. Cells were then washed three times with ice-cold PBS, and fluorescence intensities were determined on a FACSCanto II flow cytometer (BD Biosciences).

## Ligand Binding assays

Ligand binding experiments were performed on whole cells or on cell membranes obtained from transiently transfected HEK293T cells, using in both cases an HTRF binding assay as described before[11,14]. All receptor variants were subcloned into a mammalian expression vector containing an N-terminal SNAP-tag (Cisbio). For NTR1 constructs, the SNAP tag was fused to residue 43 of the receptor. For PTH1R, the SNAP tag was either fused to residue 29 or to residue 171, thus eliminating the ECD. HEK293T cells were transiently transfected with receptor constructs and were seeded at 20,000 cells per well in poly-L-lysine-coated 384-well plates (Greiner) for whole-cell binding assays or at $5 \times 10^6$ cells in 10 cm Petri dishes for membrane preparation. 48 h after transfection, cells were incubated with 50 nM SNAP-Lumi4-Tb (Cisbio) in ligand binding buffer [20 mM HEPES pH 7.5, 100 mM NaCl, 3 mM MgCl$_2$ and 0.02% (w/v) BSA] for 2 h at 37 °C. Cells were washed four times with assay buffer and used directly for whole cell ligand binding experiments, or crude cell membrane extracts were prepared as described before. Cells or 0.2–1 μg membranes per well were then incubated for 4 h on ice to measure ligand binding, containing fluorescently labeled tracer peptide together with a concentration range of unlabeled competitor peptide. For NTR1, 2 nM of HL488-NT(8–13) was used as a tracer peptide. For PTH1R, 50 nM of M-PTH(1–14)-HL647 or 20 nM of PTH'(1–34)-HL647 were used. Fluorescence intensities were measured on a Spark fluorescence plate reader (Tecan) with an excitation wavelength of 340 nm and emission wavelengths of 620 nm, 520 nm and 665 nM for Tb$^{3+}$, HiLyte Fluor 488 and HiLyte Fluor 647, respectively. The ratio of FRET-donor and -acceptor fluorescence intensities was calculated. Total binding was obtained in the absence of competitor, and nonspecific binding was determined in the presence of a 100-fold excess of unlabeled peptide. Data were normalized to the specific binding for each individual experiment and were analyzed by global fitting to a one-site heterologous competition equation.

## Signaling assays

Signaling experiments were performed on whole cells with transiently transfected HEK293T cells as described before[11,14]. Twenty-four hours after transfection, cells were washed with PBS, detached with cell dissociation buffer (Gibco) and washed again in PBS. Cells were resuspended in assay buffer [10 mM Hepes (pH 7.4), 146 mM NaCl, 1 mM CaCl$_2$, 0.5 mM MgCl$_2$, 4.2 mM KCl, 5.5 mM glucose, 50 mM LiCl, 1 mM 3-isobutyl-1-methylxanthin]. cAMP and IP1 accumulation assays were performed on white low-volume 384-well plates (Greiner) using the cAMP Tb kit and the IP-One Tb kit (both from CisBio), respectively, according to the manufacturer's protocol. For cAMP accumulation, 5000 cells were incubated with agonist at the indicated concentrations for 30 min at RT. To determine basal receptor signaling, cells were incubated in isobutylmethylxanthine (IBMX)-containing assay buffer for 30 min in the absence of ligand. For IP1 accumulation, 20,000 cells were incubated with agonist at the indicated concentrations for 2 h at

37 °C. Fluorescence intensities were measured on a Spark fluorescence plate reader (Tecan). To generate concentration-response curves, data were fitted to a three-parameter logistic equation.

## Thermostability measurements

Stability of evolved receptor variants was measured in membrane fractions of transiently transfected HEK293T cells by determining the residual receptor-bound ligand after a heat challenge of the membranes. In the case of PTH1R, the ECD (residues 1–170) was removed from the expression constructs to restrict stability measurements to the TMD. For NTR1, cells were left unmodified, whereas for PTH1R, cells were labeled with 50 nM SNAP-Lumi-4Tb before membranes were prepared as described above. Membranes were then incubated for 2–4 h on ice in ligand-binding buffer containing 20 nM of [3,11-tyrosyl-3,5-$^3$H(N)]-neurotensin (Perkin Elmer) and 500 nM of M-PTH(1–14)-HL647 for NTR1 and for PTH1R, respectively. Where indicated, 25 μM of mini-G$_s$ protein were added to the membrane fractions prior to ligand addition. Thereafter, 0.5 μg of membranes were distributed per well of a 96-well plate and heated to a specific temperature in a PCR thermocycler for 20 min. NTR1-containing membranes were then immobilized on glass fiber filters (Millipore), washed four times with binding buffer, and the residual activity of the radio-ligand was measured on a MicroBeta Plus 1450 liquid scintillation counter (Perkin Elmer). For PTH1R, residual ligand binding was determined by HTRF as described above. Data were analyzed by nonlinear regression fitting.

## Data quantification, statistical analysis and visualization

Flow cytometry data were analyzed in FlowJo software V10 (BD Biosciences). Data collection and analysis was performed in Microsoft Excel V2108. All other statistical analysis and curve fitting was performed in Prism V6.07 (GraphPad). Details of each analysis are outlined in the experimental methods section, figures, tables and figure legends of the specific experiment. Sequence alignments and snake plots were obtained from the GPCRdb[58]. Sequence analysis was performed with CLC Workbench V22.0.2. Sequence frequencies were visualized with WebLogo[59]. Protein structures were analyzed and visualized with PyMOL V2.8.2. Statistical significance of differences were determined by one-way ANOVA and Bonferroni multi-comparison.

## Statistics and reproducibility

No statistical method was used to predetermine sample size. No data were excluded from the analyses. The experiments were not randomized. The Investigators were not blinded to allocation during experiments and outcome assessment.

## Reporting summary

Further information on research design is available in the Nature Portfolio Reporting Summary linked to this article.

# Data availability

All data needed to evaluate the conclusions in the paper are present in the paper and/or in the Supplementary Information. Publicly available PDB entries used in this work: 4BUO, 4L6R). Receptor sequence data used in this paper are available from GPCRdb. Source data are provided with this paper.

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

## Acknowledgements

We thank Frank Murante and Kari Viggiani for expert technical assistance. We would also like to acknowledge Pascal Egloff for valuable inputs in the library design. This work was supported by a fellowship of the German Academy of Sciences Leopoldina (LPDS 2009-48) and a Marie Curie fellowship of the European Commission (FP7-PEOPLE-2011-IEF #299208) to C.K. and by a grant from the Schweizerische Nationalfonds (31003A_182334) to A.P.

## Author contributions

C.K., E.S.S., M.Z., and A.P. designed research; C.K., M.S., A.N., S.S., L.M., and E.G. performed research; R.S. contributed new reagents; C.K., M.S., A.N., E.S.S., M.Z., and A.P. analyzed data; C.K. and A.P. wrote the paper.

## Competing interests

M.S., S.S., L.M., E.G., M.Z. and E.S.S. are employees of Vaccinex, Inc. and own stock and/or stock options in the company. R.S. is an employee of MorphoSys AG and declares that no competing interests exist. The remaining authors declare no competing interests.
