## [Peer Review File · Nature Communications]

REVIEWER COMMENTS

Reviewer #1 (Remarks to the Author):

The authors present a method for the optimization of GPCR expression in mammalian cells. The method is comprised of 5 core steps: 1) DNA synthesis to create sequence diversity 2) packaging the GPCR library into a poxvirus 3) delivering the poxvirus to cells at low MOI 4) using FACS to isolate cells that highly express the GPCR and 5) using PCR to recover those GPCR variants of interest. Using this scheme, the authors find mutations for NTR1 and PTH1R that stabilize their expression in mammalian cells and retain their signaling capacity.

The two campaigns differed slightly. For NTR1 the authors added a mutation that abolished G-protein binding, including this in all variants. For PTH1R the authors use the wildtype sequence. The results proceed accordingly: the NTR1 variants showing higher thermostability in the absence of G-protein and PTH1R variants requiring the presence of G-protein. The investigations into NTR1 and PTH1R complement each other nicely and highlight complexities that should be considered when screening for variants in mammalian cells. The work stands as a pure methodology. While many mutations were found, novel signaling mechanisms of wildtype receptors were not uncovered.

Comments

1) Presumably this approach is used to facilitate crystallographic study. However, Cryo-EM has enabled GPCR structure determination to such a degree that the potential impact/use of this new method is limited. The authors should comment on the ideal application of this method.

2) Indeed, the thermostabilizing mutations, like the one presented in this work, can greatly change the native structure of a protein. This has even been shown specifically for PTH1R by Eric Xu and colleagues, demonstrating significant structural differences between native and thermally stabilized receptors. If the end goal is structure determination the authors should comment on this specific issue. The PTH1R cryo paper should also be cited and discussed: Zhao et al, Science 2019.

3) Directed evolution by definition requires more than one round of mutagenesis. This work presents only one round of mutagenesis. The work presented herein aligns better as a "library screen" rather than a "directed evolution campaign".

4) The authors overstate the limitations caused by lentiviruses. There exist technologies that can perform similarly to lentiviruses but with more uniform expression. For example, see Staller et al. 2022 Cell Systems, which uses a Cre based system for 1 knock in per cell at a defined genetic locus. Findlay et al. 2018 Nature, which uses CRISPR based knockin to insert a library at a defined genetic locus. Serine integrases can also be used for the same purpose, see Findlay et al. 2022 Nature Biotechnology.

5) The libraries were not sequenced and thus the true diversity and coverage of these libraries remain unclear. Diversity loss can occur from the manufacturer, during PCR amplification, cloning into a plasmid, transformation, and growth in liquid culture. Efficient bacterial transformation is necessary but not sufficient for a highly diverse library. Sequencing the final DNA library would provide a measure of diversity.

6) The authors could add more detail to make the work more accessible to their target audience. The commercial vendor that created the GPCR libraries is obscure and does not seem to have an active website. Depositing Vaccinia virus plasmid helpers would help expand access. If another poxvirus species would also work well please describe.

7) The mutational identity of variants should be made available. This was not present in the main

figures or supplement.

8) "Many of these assays are fluorescence-based and are compatible with flow cytometry applications. Integration of such sensors for receptor selection will further increase the versatility of our mammalian evolution system, as receptor evolution can be directed towards specific functional properties." Could the authors provide citations to particular assays they had in mind.

Reviewer #2 (Remarks to the Author):

In their manuscript 'A Vaccinia-based system for directed evolution of GPCRs in mammalian cells', Klenk and co-workers describe a new approach for the directed evolution of G protein-coupled receptors in mammalian cells. Together with the rational design of synthetic DNA libraries, the authors employed this Vaccinia virus-based approach to evolve two GPCRs, the neurotensin receptor 1 (NTR1) and the parathyroid 1 receptor (PTH1R). By using a NTSR1 mutant with significantly impaired G protein coupling, the authors stabilized the receptor in conformational states that show agonist peptide binding with higher affinity and increased thermostability compared to the original NTR1_R167L construct. In addition, directed evolution of a signaling active PTH1R receptor using two differently sized agonists, resulted in receptor variants with improved agonist-binding affinity and/or G protein coupling. This work provides an important new addition to the currently available set of tools for the directed evolution of GPCRs in bacteria, yeast, and mammalian cells to generate GPCR variants with improved expression, stability and signaling profiles. The manuscript is well written and illustrated and would be of great interest for the GPCR community. I do have some points that I would like to see addressed before recommending this manuscript for publication in Nature Communications:

- Several of the NTR1 variants evolved in mammalian cells show significantly enhanced expression levels in comparison to the constructs evolved in *E. coli*. In order to better understand which mutations contribute in the increased expression and stability of the NTR1 derivatives, the amino acid distribution of the clones after the two selection rounds should be shown and compared with the two previously evolved NTSR1 variants. This data would be also interesting for the structural interpretation of the described uncoupling of the allosteric connection between the orthosteric site and the intracellular G protein-binding site.
- Did the authors analyze the amino acid frequencies of diversified positions in their library to see if the library frequencies match the target values? This information would be worth including in the manuscript.
- At several positions, the authors mention that the directed evolution of the NTR1 led to a stabilization of the inactive state. Did the authors try to reverse the leucine substitution at position 167 for the selected clones in order to analyze the effect of the evolved mutations on IP1 accumulation alone? The N13 clone, e.g., seems to show unaltered IP1 accumulation in comparison to the NTR1_R167L construct.
- In line 229, the authors mention that the PTH1R variants showed an up to 13-fold increase in expression relative to the WT. While this value is listed in suppl. Table 3 for clone P34_07, the values in Fig. 3 seem to be somehow different (P34_07 shows a 6-7 fold increase in receptor expression). These values should be checked for consistency for all described clones.
- The population of high fluorescent clones in Fig. 3A is quite low after three rounds of selections, as stated by the authors. Since a lot of the selected clones show higher cAMP responses, could it be possible that this observation is caused by an increase in the internalization of the receptor derivatives rather than the instability of the cells during FACS sorting? This potential drawback should be included in the discussion.
- Similar to NTR1, it would be helpful for the interpretation of the results to provide an analysis of the mutation frequency of residues of the PTH1R after the selections. An interesting aspect would be if some of the mutations are found in the conserved sequence motifs that are known to be important for stabilizing the inactive or active state of Class B GPCRs.
- While the data in Fig. 4A provides some evidence that the evolution of the receptor resulted in the

stabilization of conformations with improved agonist affinity and decreased antagonist binding, the ligand binding curves for the PTH'(1-34) ligand should be shown as well due to the impact of the ECD interaction on the overall ligand binding affinity. This would be a fairer comparison with the ligands PTH(3-34) and IA-PTH(7-34), because both of these peptides possess the C-terminal part that interacts with the ECD, while the truncated PTH(1-14) peptide only binds to the transmembrane domain.

- Line 278 and line 388: GPCR-G protein complexes can be destabilized by both nucleotides, GDP and GTP. Therefore, GDP should be replaced by 'nucleotides' or both, GDP and GTP, should be mentioned here.

- Line 283: The evolved clones seem to be stabilized in an active conformation that increases the agonist binding affinity. Since this is not necessarily identical with the G protein-bound conformation, the authors should replace 'a G protein-bound active-state conformation' with 'an active state conformation'.

- How do the authors explain that the clone P14_12 shows a lower Emax value in comparison to the WT (57%; Fig. 3C and suppl. Table 4), but exhibits constitutive activity with a slightly but significantly higher basal activity compared to the WT? In addition, why does the clone P34_05 do not show an effect of G protein addition on the delta-pIC50 in Fig. 4B, but on the fraction of bound ligand in Fig. 4C?

- Line 313: A high capacity for G protein signaling of evolved GPCR clones can be toxic to mammalian cells. Furthermore, GPCRs that are stabilized in an active conformation can also be faster internalized due to their potential better interaction with GRKs and/or arrestins. Since these receptor variants will show lower protein levels in the plasma membrane, they will most likely be missed by using FACS sorting based on membrane-impermeable fluorescent ligands. These limitations should be included in the discussion of potential challenges for the selection of GPCR clones with improved G protein coupling or constitutive activity.

- The authors should provide more details in the method section on the synthesis of the DNA libraries.

Below, we provide a point-by-point response of the reviewers' comments. To facilitate reading, we have copied the comments in italics and we provide our response directly underneath in plain text.

REVIEWER COMMENTS

Reviewer #1 (Remarks to the Author):

The authors present a method for the optimization of GPCR expression in mammalian cells. The method is comprised of 5 core steps: 1) DNA synthesis to create sequence diversity 2) packaging the GPCR library into a poxvirus 3) delivering the poxvirus to cells at low MOI 4) using FACS to isolate cells that highly express the GPCR and 5) using PCR to recover those GPCR variants of interest. Using this scheme, the authors find mutations for NTR1 and PTH1R that stabilize their expression in mammalian cells and retain their signaling capacity.

The two campaigns differed slightly. For NTR1 the authors added a mutation that abolished G-protein binding, including this in all variants. For PTH1R the authors use the wildtype sequence. The results proceed accordingly: the NTR1 variants showing higher thermostability in the absence of G-protein and PTH1R variants requiring the presence of G-protein. The investigations into NTR1 and PTH1R complement each other nicely and highlight complexities that should be considered when screening for variants in mammalian cells. The work stands as a pure methodology. While many mutations were found, novel signaling mechanisms of wildtype receptors were not uncovered.

Comments

- 1) *Presumably this approach is used to facilitate crystallographic study. However, Cryo-EM has enabled GPCR structure determination to such a degree that the potential impact/use of this new method is limited. The authors should comment on the ideal application of this method.*

Response: Thermostabilization has often proven to be essential for crystallography of GPCRs, and many studies aiming for high-resolution rely on crystal structures. Moreover, almost all Cryo-EM GPCR structures are in the active state, since receptor-G protein complexes are required to meet the still remaining size limits of Cryo-EM. Thus, crystallography is still method of choice to obtain inactive state structures of (stabilized) receptors (though we acknowledge the recent exception using an inactive-state stabilizing nanobody demonstrated by the Skiniotis lab (*Robertson et al., Nat. Struct. Mol. Biol. 2022*)). Nonetheless, thermostabilization unequivocally has its value also for Cryo-EM studies as protein quality is one of the critical parameters for the success of any structure determination (c.f. *Zhang et al., Nat. Struct. Mol. Biol. 2021; Waltenspühl et al., Nat. Commun. 2022; Lee et al., Nature 2019*). As such, our approach will be useful for future structural discovery campaigns.

Moreover, our approach now also allows to address functional aspects of GPCRs, such as generating receptor mutants that can be used for drug screening purposes or to study functional aspects such as biased signaling in pharmacological studies. Importantly, fragment-based screening is extremely challenging, usually impossible, in cell-based assays, and the high demands on stability require the use of stable yet functional mutants. Therefore, we foresee broad applications of our method beyond structural studies, which we have already exemplified in the discussion.

- 2) *Indeed, the thermostabilizing mutations, like the one presented in this work, can greatly change the native structure of a protein. This has even been shown specifically for PTH1R by Eric Xu and colleagues, demonstrating significant structural differences between native and thermally stabilized receptors. If the end goal is structure determination the authors*

should comment on this specific issue. The PTH1R cryo paper should also be cited and discussed: Zhao et al, Science 2019.

Response: We agree that thermostabilization can have an impact on the receptor structure as certain conformational states are stabilized, and, as noted by the reviewer, for PTH1R the crystal structure (6FJ3) and the Cryo-EM structures by Zhao et al. (6NBF, 6NBH, 6NBI) reflect distinct receptor states, resulting in different conformations. However, we would like to emphasize that structures of thermostabilized receptors *per se* do not exhibit conformations that are not compatible with conformations native receptors can adopt. Yet, as stated above, our approach is not solely designed to identify thermostabilizing mutations for structural studies.

We now refer to the active state structures by Zhao et al. and the more recent ones when detailing the potential effects of single mutations of the selected PTH1R variants (c.f. response to comment 6 of reviewer 2).

- 3) *Directed evolution by definition requires more than one round of mutagenesis. This work presents only one round of mutagenesis. The work presented herein aligns better as a “library screen” rather than a “directed evolution campaign”.*

Response: We agree that in most cases directed evolution is a combination of randomization and selection, which is run iteratively in order to achieve sufficient diversity without compromising library integrity by introducing excessive mutational load at each step. However, here we have optimized library design in such a way that repetitive randomization steps were not required. Yet, variants were selected for a specific trait from a pool of variants, which is one of the key features of evolution. In contrast, screening typically describes a sequential process where mutants are first isolated and then tested individually. Thus, we believe that directed evolution describes our approach best.

- 4) *The authors overstate the limitations caused by lentiviruses. There exist technologies that can perform similarly to lentiviruses but with more uniform expression. For example, see Staller et al. 2022 Cell Systems, which uses a Cre based system for 1 knock in per cell at a defined genetic locus. Findlay et al. 2018 Nature, which uses CRISPR based knockin to insert a library at a defined genetic locus. Serine integrases can also be used for the same purpose, see Findlay et al. 2022 Nature Biotechnology.*

Response: We thank the reviewer for suggesting additional methods to generate mammalian display libraries (we believe that the last example was referring to Durrant et al., Nat. Biotech. 2022), and we have now added those to the introduction. However, as detailed in the manuscript, systems integrating into the host genome – including the above-mentioned variations – suffer from significantly lower efficacy and thus only allow replication of small libraries. Moreover, only a single gene copy is maintained in the expression host when using retro-viral systems under these conditions. This greatly limits expression levels of many proteins, notably GPCRs, and often leads in inhomogeneous expression (our own observations). In contrast, *Vaccinia* virus replicates in the cytoplasm to a defined copy number and thus enables a controlled amplification of the gene of interest with homogenous expression levels. Therefore, we believe that our approach is superior for the given application.

- 5) *The libraries were not sequenced and thus the true diversity and coverage of these libraries remain unclear. Diversity loss can occur from the manufacturer, during PCR amplification, cloning into a plasmid, transformation, and growth in liquid culture. Efficient bacterial transformation is necessary but not sufficient for a highly diverse library. Sequencing the final DNA library would provide a measure of diversity.*

Response: We thank the reviewer for this valuable comment. Unfortunately, we do not have deep-sequencing data from the final libraries. Yet, all libraries have been Sanger-sequenced after synthesis for quality control. We have now performed a thorough analysis of these data, and we added new subpanels to Supplementary Figs. 1 & 4, depicting the diversity and quality of the naïve libraries for NTR1 and PTH1R, respectively (see also response to comment 2 of reviewer 2).

- 6) *The authors could add more detail to make the work more accessible to their target audience. The commercial vendor that created the GPCR libraries is obscure and does not seem to have an active website. Depositing Vaccinia virus plasmid helpers would help expand access. If another poxvirus species would also work well please describe.*

Response: DNA libraries used in this study were custom-synthesized by Sloning, which is now part of Morphosys AG (www.morphosys.com), a large biopharmaceutical company. The details of the technology behind the DNA libraries have been published and were cited in the manuscript. Moreover, we have now added additional details to the synthesis in the methods section. At the time this study was started, the Slonomics technology to generate complex position-specific libraries was unique and thus has been the method of choice here. However, in the meantime several companies offer a variety of approaches to custom-synthesize targeted DNA libraries according to a user design, and we believe that these will work equally well in our approach.

Likewise, methods for creation of *Vaccinia* libraries by trimolecular recombination have been described in detail before and are freely accessible (e.g. *Smith et al., Meth. Mol. Biol. 2004*). While not tested, also other poxvirus strains may work for our approach. However, *Vaccinia* virus has been characterized best for its biosafety history and handling routines. Moreover, *Vaccinia* virus exhibits infection patterns that are superior to other pox viruses, which is a critical feature for sampling of large libraries.

- 7) *The mutational identity of variants should be made available. This was not present in the main figures or supplement.*

Response: We have now added new Supplementary figures 3 and 5, detailing the sequence identities of the selected clones for NTR1 and PTH1R. Moreover, we compare the distribution of mutations to known conserved positions in GPCRs and to previously evolved variants in *E. coli* in the main text (see also response to comments 1 & 6 from reviewer 2).

- 8) *“Many of these assays are fluorescence-based and are compatible with flow cytometry applications. Integration of such sensors for receptor selection will further increase the versatility of our mammalian evolution system, as receptor evolution can be directed towards specific functional properties.” Could the authors provide citations to particular assays they had in mind.*

Response: We have now specified assays in the text along with accompanying citations

Reviewer #2 (Remarks to the Author):

In their manuscript 'A Vaccinia-based system for directed evolution of GPCRs in mammalian cells', Klenk and co-workers describe a new approach for the directed evolution of G protein-coupled receptors in mammalian cells. Together with the rational design of synthetic DNA libraries, the authors employed this Vaccinia virus-based approach to evolve two GPCRs, the neurotensin receptor 1 (NTR1) and the parathyroid 1 receptor (PTH1R). By using a NTSR1 mutant with significantly impaired G protein coupling, the authors stabilized the receptor in conformational states that show agonist peptide binding with higher affinity and increased thermostability compared to the original NTR1_R167L construct. In addition, directed evolution of a signaling active PTH1R receptor using two differently sized agonists, resulted in receptor variants with improved agonist-binding affinity and/or G protein coupling. This work provides an important new addition to the currently available set of tools for the directed evolution of GPCRs in bacteria, yeast, and mammalian cells to generate GPCR variants with improved expression, stability and signaling profiles. The manuscript is well written and illustrated and would be of great interest for the GPCR community. I do have some points that I would like to see addressed before recommending this manuscript for publication in Nature Communications:

- 1) *Several of the NTR1 variants evolved in mammalian cells show significantly enhanced expression levels in comparison to the constructs evolved in E. coli. In order to better understand which mutations contribute in the increased expression and stability of the NTR1 derivatives, the amino acid distribution of the clones after the two selection rounds should be shown and compared with the two previously evolved NTSR1 variants. This data would be also interesting for the structural interpretation of the described uncoupling of the allosteric connection between the orthosteric site and the intracellular G protein-binding site.*

Response: We have now added the sequence information of the selected NTR1 variants along with mutations in NTR1 variants TM86V and L5X from previous evolution campaigns (new Supplemental Fig. 3.). Furthermore, we now describe some of these mutations in the results section. (see also response to comment 7 of reviewer 1)

- 2) *Did the authors analyze the amino acid frequencies of diversified positions in their library to see if the library frequencies match the target values? This information would be worth including in the manuscript.*

Response: We thank the reviewer for this suggestion. While we have not performed deep sequencing to obtain the full library diversity, multiple members of both libraries were Sanger-sequenced after library synthesis to determine quality and diversity. We have now added corresponding subpanels entailing the library quality to Supplemental Figs. 2 and 4 along with mentioning the results in the main text (see also response to comment 5 of reviewer 1).

- 3) *At several positions, the authors mention that the directed evolution of the NTR1 led to a stabilization of the inactive state. Did the authors try to reverse the leucine substitution at position 167 for the selected clones in order to analyze the effect of the evolved mutations on IP1 accumulation alone? The N13 clone, e.g., seems to show unaltered IP1 accumulation in comparison to the NTR1_R167L construct.*

Response: Currently, we have not performed backmutation of the DRY motif in any of the clones. However, we have shown in previous work that restoration of the DRY motif in the presence of stabilizing mutations makes the receptor able to signal, but does not fully recover receptor function, suggesting that effects of single mutations on receptor function are rather independent (c.f. *Egloff et al., PNAS 2014*). Clone N23 (we believe this is the clone the reviewer was referring to) is standing out from the panel of variants, as it indeed

has an E_{\max} comparable to the underlying NTR1-R167L mutant. Yet, it carries only one additional mutation R392C on top of the three hotspot mutations mentioned above. However, R392 is located in the unstructured C-terminus making it difficult attribute any functional role to this residue.

In this regard, we would also like to apologize for the crowded data representation in Supplementary Fig. 2, which was indeed difficult to interpret. To make it more comprehensive and at the same time more understandable, we have now split variants into two groups according to their E_{\max} values.

- 4) *In line 229, the authors mention that the PTH1R variants showed an up to 13-fold increase in expression relative to the WT. While this value is listed in suppl. Table 3 for clone P34_07, the values in Fig. 3 seem to be somehow different (P34_07 shows a 6-7 fold increase in receptor expression). These values should be checked for consistency for all described clones.*

Response: We thank the reviewer for this critical assessment of our data. Indeed, we had made a mistake in data transfer which resulted in incorrect data in Suppl. Table 3. We have now corrected the table data and the corresponding text at line 245 in the revised document.

- 5) *The population of high fluorescent clones in Fig. 3A is quite low after three rounds of selections, as stated by the authors. Since a lot of the selected clones show higher cAMP responses, could it be possible that this observation is caused by an increase in the internalization of the receptor derivatives rather than the instability of the cells during FACS sorting? This potential drawback should be included in the discussion.*

Response: Indeed, maintaining a high-expressing population during selection was quite demanding for PTH1R. As suggested by the reviewer, a possible explanation for this could be ligand-independent internalization and possibly toxic effects of mutants that exhibit high constitutive activity. We have now discussed this in more detail in the manuscript (c.f. our response to comment #11).

- 6) *Similar to NTR1, it would be helpful for the interpretation of the results to provide an analysis of the mutation frequency of residues of the PTH1R after the selections. An interesting aspect would be if some of the mutations are found in the conserved sequence motifs that are known to be important for stabilizing the inactive or active state of Class B GPCRs.*

Response: We have now added the sequence information of the selected PTH1R variants (new Supplemental Fig. 5) and have compared common mutations with previous reports (see also response to comment 7 of reviewer 1). While there are no obvious changes in conserved motifs of class B GPCRs, we find a high mutation load in areas that are in close contact with the N-terminal part of PTH peptides and that are involved in initiating receptor activation.

- 7) *While the data in Fig. 4A provides some evidence that the evolution of the receptor resulted in the stabilization of conformations with improved agonist affinity and decreased antagonist binding, the ligand binding curves for the PTH'(1-34) ligand should be shown as well due to the impact of the ECD interaction on the overall ligand binding affinity. This would be a fairer comparison with the ligands PTH(3-34) and IA-PTH(7-34), because both of these peptides possess the C-terminal part that interacts with the ECD, while the truncated PTH(1-14) peptide only binds to the transmembrane domain.*

Response: As requested by the reviewer, we have now included a new subpanel to Figure 4A showing the inhibition binding curves with PTH(1-34) to the respective PTH1R variants.

- 8) *Line 278 and line 388: GPCR-G protein complexes can be destabilized by both nucleotides, GDP and GTP. Therefore, GDP should be replaced by 'nucleotides' or both, GDP and GTP, should be mentioned here.*

Response: We thank the reviewer for this comment and have changed the wording from 'GDP' to 'nucleotides' in the manuscript accordingly.

- 9) *Line 283: The evolved clones seem to be stabilized in an active conformation that increases the agonist binding affinity. Since this is not necessarily identical with the G protein-bound conformation, the authors should replace 'a G protein-bound active-state conformation' with 'an active state conformation'.*

Response: We thank the reviewer for this comment and have changed the manuscript accordingly.

- 10) *How do the authors explain that the clone P14_12 shows a lower Emax value in comparison to the WT (57%; Fig. 3C and suppl. Table 4), but exhibits constitutive activity with a slightly but significantly higher basal activity compared to the WT? In addition, why does the clone P34_05 do not show an effect of G protein addition on the delta-pIC50 in Fig. 4B, but on the fraction of bound ligand in Fig. 4C?*

Response: We believe that in the case of variant P14_12 basal activity and ligand-induced receptor activity may be partly uncoupled. This can happen, when e.g. important allosteric networks are broken that link the orthosteric binding site to the G protein interface, and thus the ligand cannot activate the receptor to full extent. Such effects of mutations are not uncommon and have also been observed by us and others before (e.g. *Egloff et al., PNAS 2014; Krumm et al., Sci. Rep. 2016; Urizar et al., JBC 2005*).

Indeed, variant P34_05 shows an increase in ligand occupancy upon addition of G protein. As this variant has by far the highest apparent affinity for G protein, we speculate that any changes in ligand affinity by G protein may be subtle and thus may be obscured by experimental variance.

- 11) *Line 313: A high capacity for G protein signaling of evolved GPCR clones can be toxic to mammalian cells. Furthermore, GPCRs that are stabilized in an active conformation can also be faster internalized due to their potential better interaction with GRKs and/or arrestins. Since these receptor variants will show lower protein levels in the plasma membrane, they will most likely be missed by using FACS sorting based on membrane-impermeable fluorescent ligands. These limitations should be included in the discussion of potential challenges for the selection of GPCR clones with improved G protein coupling or constitutive activity.*

Response: We thank the reviewer for this valuable comment. Indeed, this is a possibility when receptors exhibit high levels of constitutive activity (see also our response to comment 5). Receptor variants that only signal in response to ligand binding, however, will most likely not be affected as we apply ligand only very shortly before the FACS selection. In fact, in such a case, co-internalization of fluorescent ligands would make FACS-based selection even more robust, as ligand dissociation can be neglected. Also, toxic effects from increased signaling after ligand addition will probably not affect selection, as cells are only in short contact with the labelled ligand prior to the FACS procedure. We have discussed this issue in more detail in the main text.

12) *The authors should provide more details in the method section on the synthesis of the DNA libraries.*

Response: We have now included a more detailed description of the library generation using Slonomics technology (see also response to comment 6 of reviewer 1).

References:

Egloff, P. *et al.* Structure of signaling-competent neurotensin receptor 1 obtained by directed evolution in *Escherichia coli*. *Proc. Natl. Acad. Sci. U. S. A.* **111**, E655–62 (2014).

Krumm, B. E. *et al.* Structure and dynamics of a constitutively active neurotensin receptor. *Sci. Rep.* **6**, 185 (2016).

Lee, Y. *et al.* Molecular basis of β -arrestin coupling to formoterol-bound β 1-adrenoceptor. *Nature* **583**, 862–866 (2020).

Robertson, M. J. *et al.* Structure determination of inactive-state GPCRs with a universal nanobody. *Nat. Struct. Mol. Biol.* **29**, 1188–1195 (2022).

Smith, E. S., Shi, S. & Zauderer, M. Construction of cDNA libraries in vaccinia virus. *Methods Mol Biol* **269**, 65–76 (2004).

Urizar, E. *et al.* An activation switch in the rhodopsin family of G protein-coupled receptors: the thyrotropin receptor *J Biol Chem.* **280**, 17135-17141 (2005).

Waltenspühl, Y. *et al.* Structural basis for the activation and ligand recognition of the human oxytocin receptor. *Nat. Commun.* **13**, 4153 (2022).

Zhang, M. *et al.* Cryo-EM structure of an activated GPCR-G protein complex in lipid nanodiscs. *Nat. Struct. Mol. Biol.* **28**, 258–267 (2021).

REVIEWERS' COMMENTS

Reviewer #1 (Remarks to the Author):

The authors have adequately addressed my concerns.

Reviewer #2 (Remarks to the Author):

The authors sufficiently addressed all my concerns and comments and significantly improved the manuscript by including the mutation frequencies of the selected clones. I do not have any further comments and can now fully support the publication of the manuscript by Klenk and colleagues in Nature Communications.